# QWHA: Quantization-Aware Walsh-Hadamard Adaptation for Parameter-Efficient Fine-Tuning on Large Language Models

**Hyesung Jeon**[1*], **Seojune Lee**[1*], **Beomseok Kang**[1], **Yulhwa Kim**[2], **Jae-Joon Kim**[1†]

[1]Seoul National University, [2]Sungkyunkwan University

`{hjeon2k, leeseojune, beomseok, kimjaejoon}@snu.ac.kr`,
`{yulhwakim}@skku.edu`

## Abstract

The demand for efficient deployment of large language models (LLMs) has driven interest in quantization, which reduces inference cost, and parameter-efficient fine-tuning (PEFT), which lowers training overhead. This motivated the development of quantization-aware PEFT to produce accurate yet efficient quantized models. In this setting, reducing quantization error prior to fine-tuning is crucial for achieving high model accuracy. However, existing methods that rely on low-rank adaptation suffer from limited representational capacity. Recent Fourier-related transform (FT)-based adapters offer greater representational power than low-rank adapters, but their direct integration into quantized models often results in ineffective error reduction and increased computational overhead. To overcome these limitations, we propose QWHA, a method that integrates FT-based adapters into quantized models by employing the Walsh-Hadamard Transform (WHT) as the transform kernel, together with a novel adapter initialization scheme incorporating adaptive parameter selection and value refinement. We demonstrate that QWHA effectively mitigates quantization errors while facilitating fine-tuning, and that its design substantially reduces computational cost. Experimental results show that QWHA consistently outperforms baselines in low-bit quantization accuracy and achieves significant training speedups over existing FT-based adapters. The code is publicly available at `https://github.com/vantaa89/qwha`.

## 1 Introduction

Fine-tuning enables large language models (LLMs) to generalize beyond their pre-training, allowing adaptation to various domains (Wei et al., 2022; Liu et al., 2023; Qin et al., 2024; DeepSeek-AI et al., 2025). While full fine-tuning yields superior accuracy, it often incurs significant overhead due to the extensive computations required to update all the trainable model parameters (Loshchilov & Hutter, 2017; Zhu et al., 2025). Parameter-efficient fine-tuning (PEFT) addresses this issue by optimizing only a small subset of the parameters while leaving most of them frozen (Li & Liang, 2021; Liu et al., 2022; Hu et al., 2022; Liu et al., 2024; Kopiczko et al., 2024). Beyond reducing training overhead, recent studies have shown that combining PEFT with model compression techniques can enhance inference efficiency at the same time (Dettmers et al., 2023). Among these techniques, quantization, which lowers the bit precision of model parameters, has gained particular attention due to its robustness against accuracy degradation under high compression ratios (Frantar et al., 2023; Lin et al., 2024; Dettmers et al., 2024; Kim et al., 2024b; Shao et al., 2024; Ashkboos et al., 2024; Zhang et al., 2024; Liu et al., 2025). Consequently, quantization-aware PEFT (QA-PEFT) has been widely explored as a promising approach for efficient adaptation and inference in LLMs.

Prior works on QA-PEFT typically relied on low-rank adaptation (LoRA) (Li et al., 2024; Guo et al., 2024; Kim et al., 2024a; Liao et al., 2024; Deng et al., 2025). In contrast, for standard PEFT, several alternatives to LoRA have recently been proposed to address the representational limitations of low-rank structures. In particular, Fourier-related transform (FT)-based adapters have emerged as strong alternatives. They train a sparse set of coefficients to represent weight updates in the transform domain, offering superior representational capacity (Gao et al., 2024b; Du et al., 2025; Shen et al.,

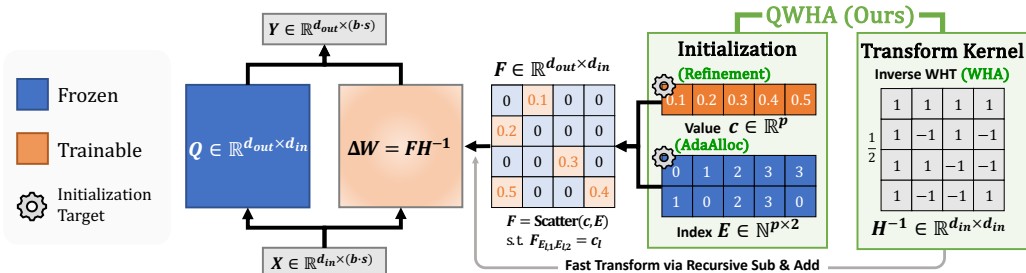

Figure 1: Overview of Quantization-aware Walsh-Hadamard Adaptation (**QWHA**). The weight update from QWHA is formulated as $\Delta W = FH^{-1}$, where $H$ is a predefined Walsh-Hadamard transform (WHT) matrix and $F$ is a trainable sparse coefficient matrix consisting of values $c$ and their indices $E$. The multiplication $FH^{-1}$ indicates the expansion of learned coefficients (*i.e.*, $c$), over the transform basis (*i.e.*, columns of $H^{-1}$). Note that, the coefficients $c$ are the only trainable parameters, and $H$ remains constant. Our key contributions are in the adoption of WHT into the adapter (**WHA**) and their initialization, particularly $E$ (**AdaAlloc**) and $c$ (**Refinement**).

Table 1: Comparison of adapter types and parameter selection strategies. Adapter types include low-rank adapters (LoRA), recent FT-based adapters (DCA and DHA), and our proposed adapter (**WHA**). Strategies to determine parameter location $E$ in $F$ include magnitude-based selection, random uniform selection, training via reparameterization, and our proposed method (**AdaAlloc**).

| | Adapter Type | | | | Parameter Selection Strategy | | | |
|---|---|---|---|---|---|---|---|---|
| Ability Factors | LoRA | DCA | DHA | **WHA** | Magnitude | Random | Trainable | **AdaAlloc** |
| Fine-tuning | ↓ | ↑ | ↑ | ↑ | ↓ | ↑ | ↑ | ↑ |
| Quantization Error Reduction | ↓ | ↓ | ↓ | ↑ | ↑ | ↓ | ↓ | ↑ |

2025). However, our observations show that directly applying FT-based adapters to quantized models often yields worse performance than LoRA-based methods specifically designed for QA-PEFT. This highlights the importance of explicit consideration for quantization effects when fine-tuning quantized models. LoRA-based methods adopt quantization-aware initialization strategies that compensate for the errors between full- and low-precision weights using low-rank approximation with the adapters prior to fine-tuning. However, applying such initialization in FT-based adapters is nontrivial, as identifying the optimal sparse set of parameters and their values to approximate a given matrix is an NP-hard problem (Natarajan, 1995). Moreover, the choice of transform type becomes an additional design consideration. This raises a research question: how to effectively exploit FT-based adapters in QA-PEFT. To the best of our knowledge, neither FT-based adapters nor their initialization techniques have been explored in the context of QA-PEFT.

In this paper, we present QWHA, a novel QA-PEFT method that introduces a FT-based adapter together with a quantization-aware initialization scheme, as illustrated in Figure 1 and Table 1. We adopt WHT in our adapter design (**WHA**), inspired by its high-fidelity reconstruction ability in the spectral domain, to effectively compensate for quantization errors (Hedayat, 1978). In addition, the WHT kernel consists solely of ±1 elements, enabling efficient computations using only additions and subtractions, thereby eliminating matrix multiplications (Dao-AILab, 2024). We further reduce computation by applying a single transform in the adapter, unlike conventional FT-based adapters that apply two transforms. For quantization-aware adapter initialization, we develop a tractable solution that first selects parameter locations $E$ and then assigns their values $c$. We introduce a channel-wise parameter allocation scheme that guarantees a lower bound on the number of parameters per channel to facilitate fine-tuning while allocating more parameters to channels with larger quantization errors, and then select the highest-magnitude coefficients within each channel to effectively reduce quantization error (**AdaAlloc**). Finally, we refine the selected parameter values, thereby enabling substantial reduction of quantization error (**Refinement**). We theoretically analyze the superior representation capacity of our proposed adapter and empirically validate the benefits of our adapter design and initialization method across diverse datasets and models.

## 2 BACKGROUND

### 2.1 LLM QUANTIZATION

LLM quantization is a key technique for improving inference efficiency by reducing the memory bottleneck caused by model weights through lowering their bit precision (Frantar et al., 2023), typically expressed by the following equation:

$$\tilde{\boldsymbol{W}}_Q = \text{clamp}\left(\text{round}\left(\frac{\boldsymbol{W}_0}{s}\right) - z, 0, 2^n - 1\right) \quad \boldsymbol{W}_Q = (\tilde{\boldsymbol{W}}_Q + z) \times s \tag{1}$$

Here, $\boldsymbol{W}_0$ denotes the pre-trained weight matrix, while $\tilde{\boldsymbol{W}}_Q$ and $\boldsymbol{W}_Q$ represent the quantized integer weights and the corresponding dequantized weights, respectively. $s$ and $z$ are quantization scales and integer zero-points. Clamping is applied to the rounded and shifted value within the range $0$ to $2^n - 1$, where $n$ is the bit-width.

LLMs generally contain outliers, a small fraction of weights that are exceptionally large compared to the main distribution, and LLM quantization is highly sensitive to these outliers (Dettmers et al., 2024; Kim et al., 2024b; Tseng et al., 2024; An et al., 2025). These outliers induce corresponding outliers in the quantization error. Most quantization errors $\Delta \boldsymbol{W}_Q = \boldsymbol{W}_0 - \boldsymbol{W}_Q$ are bounded within a small range (*e.g.*, $[-\frac{s}{2}, \frac{s}{2})$), since most weights within the clamping range are mapped to the nearest quantization level. In contrast, for outliers, the quantization error is defined as the difference between the original large weight and the clamping boundary values, resulting in extremely large errors that lead to significant accuracy degradation. Thus, reducing outlier-induced error is critical, and recent post-training quantization techniques for LLMs focus on mitigating these errors to preserve model accuracy (Dettmers et al., 2024; Kim et al., 2024b; Shao et al., 2024; Tseng et al., 2024; Zhang et al., 2024). Details on the distribution of quantization errors are presented in Appendix A.

### 2.2 QUANTIZATION-AWARE PEFT

A typical quantization-aware PEFT (QA-PEFT) adopts LoRA (Hu et al., 2022), which injects a pair of low-rank matrices into linear layers to approximate the weight updates $\Delta \boldsymbol{W}$ as follows:

$$\boldsymbol{Y} = (\boldsymbol{W}_Q + \Delta \boldsymbol{W})\boldsymbol{X} \quad s.t. \quad \Delta \boldsymbol{W} = \boldsymbol{B}\boldsymbol{A} \tag{2}$$

Here, $\boldsymbol{A} \in \mathbb{R}^{r \times d_{\text{in}}}$ and $\boldsymbol{B} \in \mathbb{R}^{d_{\text{out}} \times r}$ are low-rank adapters, fine-tuned instead of frozen quantized weight $\boldsymbol{W}_Q \in \mathbb{R}^{d_{\text{out}} \times d_{\text{in}}}$, where $\boldsymbol{X} \in \mathbb{R}^{d_{\text{in}} \times (b \times s)}$ is the activation matrix with batch size $b$ and sequence length $s$. Since there is no prior information about the weight updates before fine-tuning, LoRA typically initializes $\boldsymbol{A}$ as a random matrix and $\boldsymbol{B}$ as a zero matrix. In QA-PEFT, however, initializing the adapters to minimize quantization error prior to fine-tuning plays a crucial role in accuracy. Early approaches addressed this by reconstructing quantization errors via singular value decomposition (SVD) to initialize low-rank adapters (Li et al., 2024; Guo et al., 2024). More recent works, such as RA-LoRA (Kim et al., 2024a) and CLoQ (Deng et al., 2025), adopt advanced decomposition strategies and improved calibration to further mitigate this limitation. However, existing QA-PEFT methods remain restricted to LoRA, and no prior studies have explored the use of other advanced adapters for QA-PEFT, which will be discussed in the next section.

### 2.3 FOURIER TRANSFORM-BASED ADAPTERS

Sparse adapters have recently emerged as a strong alternative to various low-rank adapters (Bhardwaj et al., 2024; Gao et al., 2024b; Shen et al., 2025; Du et al., 2025). SHiRA (Bhardwaj et al., 2024) proposes directly updating a sparse subset of the weight matrix, enabling multi-adapter fusion. More recent methods adopt Fourier-related Transforms (FT) to represent the weight update $\Delta \boldsymbol{W}$ in the spectral domain by applying transforms along both the rows and columns of the matrix as follows:

$$\boldsymbol{F} = \boldsymbol{H}'\Delta \boldsymbol{W}\boldsymbol{H} \implies \Delta \boldsymbol{W} = \boldsymbol{H}'^{-1}\boldsymbol{F}\boldsymbol{H}^{-1} \tag{3}$$

Here, $\boldsymbol{H} \in \mathbb{R}^{d_{\text{in}} \times d_{\text{in}}}$ and $\boldsymbol{H}' \in \mathbb{R}^{d_{\text{out}} \times d_{\text{out}}}$ are the orthonormal transform kernels. Prior works on these FT-based adapters have primarily focused on identifying suitable transform kernels. FourierFT (Gao et al., 2024b) employs the discrete Fourier transform (DFT), while LoCA (Du et al., 2025) replaces the DFT with the discrete cosine transform (DCT) to avoid discarding imaginary

components. SSH (Shen et al., 2025) instead leverages the discrete Hartley transform (DHT) for the same purpose. As these kernels are composed of sinusoidal functions, $\boldsymbol{F}$ corresponds to the coefficients of the frequency components, which collectively represent $\Delta \boldsymbol{W}$. We denote DCT and DHT-based adapters as DCA and DHA throughout the paper.

Since the transform kernels are fixed matrices, $\boldsymbol{F}$ is the only learnable parameter during fine-tuning. To reduce the number of trainable parameters, $\boldsymbol{F}$ is treated as a sparse matrix. Specifically, $\boldsymbol{F} = \mathrm{Scatter}(\boldsymbol{c}, \boldsymbol{E})$ is constructed from a value vector $\boldsymbol{c} \in \mathbb{R}^p$ and an index list $\boldsymbol{E} \in \mathbb{N}^{p \times 2}$, where $\mathrm{Scatter}$ assigns $\boldsymbol{F}_{(E_{l,1}, E_{l,2})} = c_l$ for $0 \le l \le p - 1$, with all other entries fixed to zero throughout training and inference. At the initialization stage, since there is no information on $\Delta \boldsymbol{W}$, previous works generally select the locations $\boldsymbol{E}$ randomly and the values of the spectral coefficients $\boldsymbol{c}$ are initialized to zero (Gao et al., 2024b; Du et al., 2025). SSH (Shen et al., 2025) proposes an advanced parameter selection strategy under the assumption that the frequency patterns of pre-trained and fine-tuned weights are similar. It first transforms the pre-trained weights and selects half of the positions with the largest spectral coefficients, while the remaining half are chosen randomly.

Overall, previous works demonstrate that FT-based adapters achieve superior accuracy improvements in full-precision fine-tuning compared to low-rank adapters. However, their advantages over low-rank adapters have only been empirically demonstrated, without theoretical justification. In addition, transforms within FT-based adapters incur heavy computational overhead ($\boldsymbol{H}$ and $\boldsymbol{H}'$ in Equation 3). Moreover, their application to QA-PEFT, particularly with initialization strategies that reconstruct quantization error, has not yet been explored.

## 3 METHODOLOGY

In this section, we present our proposed method, **QWHA** (**Q**uantization-Aware **W**alsh-**H**adamard **A**daptation). First, we present the formulation of our proposed WHT-based adapter. Next, we analyze the key component that enables FT-based adapters to achieve greater representational capacity than low-rank adapters, and demonstrate why WHA, in particular, excels at mitigating quantization error during adapter initialization. Finally, we introduce a parameter initialization strategy that reduces quantization error and enhances fine-tuning capability. Note that the experiments in this section use the 4-bit quantized LLaMA-3.2-3B model, with the total number of trainable parameters $P(r) = \sum\limits_{l \in \text{layers}} (d_{l,\text{in}} + d_{l,\text{out}}) \times r$ fixed by setting $r = 64$ across all adapters.

### 3.1 QA-PEFT ADAPTER DESIGN

**WHT-based Adapter (WHA)** We design our proposed adapter by constructing the weight update as the transformation of a sparse matrix $\boldsymbol{F}$ through an orthogonal transform $\boldsymbol{H}^{-1}$. Specifically, we adopt the WHT (Hedayat, 1978; Kunz, 1979), a particular instance of the FT whose kernel consists only of $\pm 1$ entries, for the transform $\boldsymbol{H}$ (details on WHT and other FT kernels are provided in Appendix B.1). Accordingly, our adapter is formulated as follows:

$$\boldsymbol{Y} = (\boldsymbol{W}_Q + \Delta \boldsymbol{W}) \boldsymbol{X} \quad \text{s.t.} \quad \Delta \boldsymbol{W} = \boldsymbol{F} \boldsymbol{H}^{-1}. \tag{4}$$

The advantages of our adapter design are discussed in the following paragraphs.

**Full-Rank Adapter.** FT-based adapters exhibit greater representational capability than LoRA variants because they offer higher rank capacity given the same number of parameters. The representational power of low-rank adapters is strictly bounded by their inner dimension $r$ (Equation 2). In contrast, since the transform kernels in FT-based adapters are orthogonal and therefore full-rank, the rank of the adapter depends solely on the sparse matrix $\boldsymbol{F}$ (Equation 3 and 4). Given that nonzero parameters are selected uniformly at random, if both rows and columns receive more than two parameters on average, then $\boldsymbol{F}$ achieves full rank $r_{\max} = \min(d_{\text{in}}, d_{\text{out}})$ with high probability (Coja-Oghlan et al., 2020). Since our adapter initialization in Section 3.2 assigns at least a few elements to each channel and selects parameters independently per channel, the full-rank conditions are satisfied. Details of this condition are provided in Appendix B.2. Figure 2(a) presents the empirical analysis of the rank of adapter weights, normalized by the maximum achievable rank $r_{\max}$ and averaged across layers. While LoRA achieves less than 6.3% of the normalized rank, FT-based adapters are nearly full-rank. Hence, our proposed WHA exhibits high representational capacity.

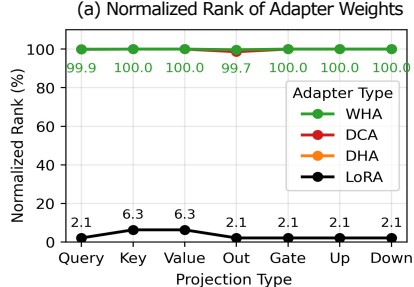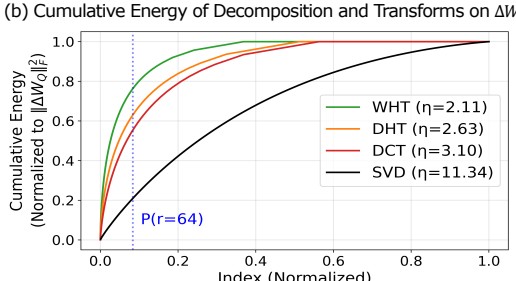

Figure 2: (a) Comparison of rank in weight updates between low-rank and FT-based adapters across linear layers. (b) Cumulative distribution of $\ell_2$ norm of singular values and transform coefficients with Pareto hill index $\eta$ for the quantization error $\Delta \boldsymbol{W}_Q$ in the 14$^{\text{th}}$-layer Value projection. The vertical blue line indicates a point where the adapters have the same number of parameters.

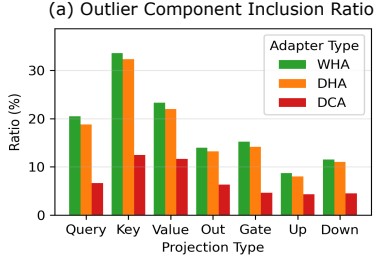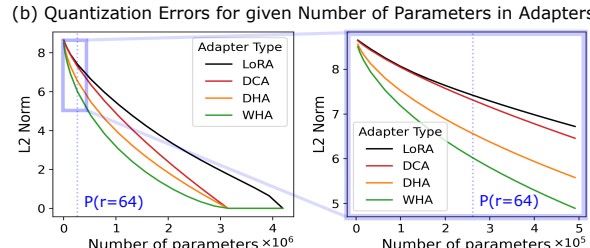

Figure 3: (a) Average coverage of outlier components within the selected parameters. (b) $\ell_2$ norm of the layer output error after initialization on the 14$^{\text{th}}$-layer Key projection. The vertical blue lines indicate points where the adapters have the same number of parameters.

**Single transform.** Conventional FT-based adapters apply transforms to both the input and output dimensions of the sparse matrix $\boldsymbol{F}$ as denoted in Equation 3. However, we find no clear advantage of this approach over a single transform in the context of quantization. Since quantization errors are defined group-wise within each output channel, the channels can be treated as independent, and multiple transforms do not improve the representational capacity (*i.e.*, rank) of the adapter. Therefore, to avoid unnecessary operations, we design WHA to perform a single transform as described in Equation 4.

**Benefits of WHT over other transforms.** As discussed in Section 2.1, quantization errors exhibit heavy-tailed outliers. For QA-PEFT, where mitigating such errors is crucial, the adapter must capture the outlier structure with a small number of parameters, as in the case of sparse adapters using the sparse matrix $\boldsymbol{F}$. We strategically adopt the WHT for our adapter design to effectively capture such outliers (Hedayat, 1978; Kunz, 1979). The WHT kernel consists only of $\pm 1$ entries, and its basis functions are square-wave patterns with sharp transitions. In contrast, prior FT-based adapters adopt DCT or DHT, whose sinusoidal bases exhibit smooth transitions. This structural difference makes the WHT better aligned with abrupt changes such as outlier values. Therefore, WHT inherently provides a more compact coefficient representation of quantization errors compared to DCT or DHT. We empirically demonstrate this by analyzing the cumulative energy in adapter parameters (Figure 2(b)), defined as the $\ell_2$ norm of coefficients from the transform of $\Delta \boldsymbol{W}_Q$ in FT-based adapters, and the $\ell_2$ norm of singular values of $\Delta \boldsymbol{W}_Q$ in low-rank adapters. Both coefficients and singular values follow a Pareto-like distribution (see Appendix B.3), which can be characterized by the Pareto hill index $\eta$, where a smaller $\eta$ indicates a sharper distribution (Arnold, 1983). Since the total cumulative energy equals $\|\Delta \boldsymbol{W}_Q\|_F^2$, the fastest convergence curve of WHT, with the smallest $\eta$, demonstrates that it concentrates the largest portion of energy within a small number of coefficients, enabling accurate reconstruction with a limited number of parameters. As a result, WHA effectively compensates for quantization errors, particularly large-magnitude ones from salient weight channels, as shown empirically in Figure 3. For a fair comparison, we use the same parameter ini-

---

**Algorithm 1** QWHA Initialization Algorithm

---

**Require:** Weight quantization error $\Delta \boldsymbol{W}_Q \in \mathbb{R}^{d_{\text{out}} \times d_{\text{in}}}$, Activation $\boldsymbol{X} \in \mathbb{R}^{d_{\text{in}} \times (b \cdot s)}$, WHT matrix $H$
**Require:** Budget $p$, Accumulated budget $\tilde{p}$, channel-wise budget $(p_0, \dots, p_{d_{\text{out}}-1}) \in \mathbb{N}^{d_{\text{out}}}$
**Require:** Parameter value vector $\boldsymbol{c} \in \mathbb{R}^p$, index list $\boldsymbol{E} \in \mathbb{N}^{p \times 2}$

   Initialize $\tilde{p}, \boldsymbol{c}, \boldsymbol{E} \leftarrow \boldsymbol{0}$
   Set $\boldsymbol{R} \leftarrow \boldsymbol{U}\boldsymbol{\Sigma}^{1/2} \leftarrow \boldsymbol{U}\boldsymbol{\Sigma}\boldsymbol{U}^\top := \text{SVD}(\boldsymbol{X}\boldsymbol{X}^\top)$
   Set $(p_0, \dots, p_{d_{\text{out}}-1}) \leftarrow \text{AdaAlloc}(p, \Delta\boldsymbol{W}_Q), \boldsymbol{B} \leftarrow \boldsymbol{H}^{-1}\boldsymbol{R}$       ▷ Parameter budget allocation
   **for** $i = 0$ to $d_{\text{out}} - 1$ **do**
      Set $\boldsymbol{v} \leftarrow (\Delta\boldsymbol{W}_Q)_{i,:}\boldsymbol{R}$
      Set $\boldsymbol{E}_{\tilde{p}, \dots, \tilde{p}+p_i-1} \leftarrow \text{TopK}_{p_i}^{\text{Index}}(\boldsymbol{v}\boldsymbol{B}^{-1})$       ▷ Channel-wise parameter selection
      Set $\boldsymbol{B}' \leftarrow \boldsymbol{B}_{(i_1, \dots, i_{p_i}),:}$
      Set $\boldsymbol{c}_{\tilde{p}, \dots, \tilde{p}+p_i-1} \leftarrow \boldsymbol{v}\boldsymbol{B}'^\top(\boldsymbol{B}'\boldsymbol{B}'^\top)^{-1}$       ▷ Value refinement
      Accumulate $\tilde{p} \leftarrow \tilde{p} + p_i$
   **end for**
   Update $\boldsymbol{F} \in \mathbb{R}^{d_{\text{out}} \times d_{\text{in}}} \leftarrow \boldsymbol{c}, \boldsymbol{E}$

---

tialization method described in Section 3.2. We define outlier coverage as the ratio of the $\ell_1$ sum of coefficients captured by the selected parameter locations to that of all coefficients corresponding to the top 10% magnitude outliers of $\Delta\boldsymbol{W}_Q$.

## 3.2 QUANTIZATION-AWARE ADAPTER INITIALIZATION

**Objective Function.** Our goal in initializing WHA is to minimize the layer output error $(\Delta\boldsymbol{W}_Q\boldsymbol{X})$ caused by weight quantization, using a coefficient matrix $\boldsymbol{F}$ with $p$ non-zero elements. Formally, the objective is given by:

$$\underset{\boldsymbol{c}, \boldsymbol{E}}{\arg\min} \|\Delta\boldsymbol{W}_Q\boldsymbol{X} - \boldsymbol{F}\boldsymbol{H}^{-1}\boldsymbol{X}\|_F^2 \tag{5}$$

where $\|\cdot\|_F$ denotes Frobenius norm. Following the reduction procedure used in Frantar et al. (2023) and Deng et al. (2025), this reduces to:

$$\underset{\boldsymbol{c}, \boldsymbol{E}}{\arg\min} \|\Delta\boldsymbol{W}_Q\boldsymbol{R} - \boldsymbol{F}\boldsymbol{H}^{-1}\boldsymbol{R}\|_F^2 \tag{6}$$

Here, $\boldsymbol{R} = \boldsymbol{U}\boldsymbol{\Sigma}^{1/2}$ is the invertible square root of the Hessian matrix attained by SVD as $\boldsymbol{X}\boldsymbol{X}^\top = \boldsymbol{U}\boldsymbol{\Sigma}\boldsymbol{U}^\top$. A detailed derivation on this reduction is provided in Appendix C.1. As we aim to find a sparse $\boldsymbol{F}(\boldsymbol{c}, \boldsymbol{E})$ that minimizes Equation 6, it constitutes an NP-hard sparse approximation problem (SAP) (Natarajan, 1995). To make this problem more tractable, we decompose it into two subproblems: first, parameter selection to determine the locations of the nonzero elements to fine-tune ($\boldsymbol{E}$); and second, value refinement to optimize the values of the selected positions ($\boldsymbol{c}$).

**Parameter Selection with AdaAlloc.** Given a number of parameter (budget) $p$ for a layer, a naive selection method to reduce quantization error is to choose the $p$ largest-magnitude elements from the dense solution $\Delta\boldsymbol{W}_Q\boldsymbol{H}$ of Equation 6. However, since large-magnitude coefficients are often clustered in a few channels containing outliers, parameters become overly concentrated in a small number of channels. As a result, magnitude-based selection yields a low-rank $\boldsymbol{F}$, degrading fine-tuning capability. Conventional methods prevent this rank reduction by incorporating random selection. For example, LoCA initializes parameter locations randomly and then optimizes these locations during fine-tuning. Thus, from the perspective of initialization, LoCA is equivalent to random selection at this stage. Additionally, SSH allocates half of the parameters randomly, while it selects the other half based on magnitude. However, these randomness-based approaches result in high layer output error because they fail to capture the parameters critical for reducing the error. To construct a sparse $\boldsymbol{F}$ that is high-rank and minimizes initialization error, we first allocate the parameter budget adaptively across output channels in proportion to their activation error magnitudes:

$$p_i \leftarrow \left\lfloor p \cdot \frac{\|(\Delta\boldsymbol{W}_Q\boldsymbol{X})_{i,:}\|_F^t}{\sum_{j=1}^{d_{\text{out}}} \|(\Delta\boldsymbol{W}_Q\boldsymbol{X})_{j,:}\|_F^t} \right\rfloor, \tag{7}$$

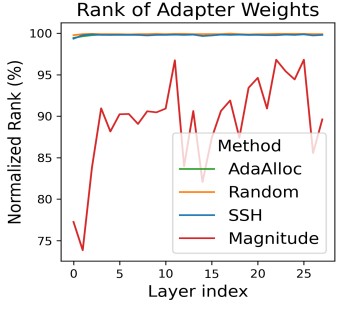

Figure 4: Rank of adapter weights for each parameter selection methods.

Table 2: Layer output error ($\ell_2$ norm, scaled by $1 \times 10^3$) after initialization. 'None' denotes the error before initialization.

| Method | None | Random | SSH | Magnitude | **AdaAlloc** |
|---|---|---|---|---|---|
| Query | 13.84 | 10.55 | 6.99 | 5.95 | 5.11 |
| Key | 0.54 | 0.43 | 0.30 | 0.25 | 0.27 |
| Value | 28.08 | 22.98 | 17.38 | 15.10 | 14.92 |
| Out | 4.66 | 3.70 | 2.70 | 2.24 | 2.01 |
| Gate | 1.88 | 1.57 | 1.25 | 1.04 | 1.13 |
| Up | 25.76 | 23.05 | 19.85 | 16.52 | 17.97 |
| Down | 21.36 | 19.21 | 16.96 | 14.00 | 15.25 |
| **Average** | 7.21 | 5.96 | 4.57 | 3.82 | 3.86 |

where $t$ is a temperature hyperparameter controlling allocation sharpness. Because the parameter budget must be an integer, we apply the floor operation, which may leave fewer than $d_{\text{out}}$ parameters unassigned. These remainders are distributed to the output channels with the smallest allocations to ensure $\sum_{i=1}^{d_{\text{out}}} p_i = p$. Since all output channels receive parameter budgets proportional to their errors, $\boldsymbol{F}$ maintains full rank, while allocating more parameters to important channels with higher quantization error. Next, within the budget of each output channel, we select parameters based on magnitude to effectively reduce the error. We compare the rank and layer output error of previous selection methods and AdaAlloc, as shown in Figure 4 and Table 2. For a fair comparison, all selection methods use the same value assignment method discussed in the next paragraph. AdaAlloc is the only parameter selection method that simultaneously achieves a nearly full-rank $\boldsymbol{F}$ and maintains low layer output error. Examples of the selected parameters are provided in Appendix C.2.

**Value Refinement.** To assign each parameter a value that effectively reduces layer output error, we solve the channel-wise SAP derived from Equation 6 for each $i^{\text{th}}$ output channel with a given parameter budget $p_i$:

$$\min_{\boldsymbol{x}} \|\boldsymbol{v} - \boldsymbol{x}\boldsymbol{B}\|_2^2, \quad \text{where} \quad \boldsymbol{v} = (\Delta\boldsymbol{W}_Q)_{i,:}\boldsymbol{R}, \quad \boldsymbol{B} = \boldsymbol{H}^{-1}\boldsymbol{R}. \tag{8}$$

Here, $\boldsymbol{x}$ is the $i^{\text{th}}$ row of $\boldsymbol{F}$, constrained to have $p_i$ non-zero elements. We first select the $p_i$ largest-magnitude entries from the channel-wise dense solution $\boldsymbol{x}_0 = \boldsymbol{v}\boldsymbol{B}^{-1} = (\Delta\boldsymbol{W}_Q\boldsymbol{H})_{i,:}$. Next, rather than directly reusing the values from a dense solution, we refine them to minimize layer output error. Specifically, we re-project $\boldsymbol{v}$ onto the rows of $\boldsymbol{B}$ corresponding to the selected indices, denoted as $\boldsymbol{B}' \in \mathbb{R}^{p_i \times d_{\text{in}}}$, which serve as the most relevant basis vectors:

$$\boldsymbol{x}^* = \boldsymbol{v}\boldsymbol{B}'^{\top}(\boldsymbol{B}'\boldsymbol{B}'^{\top})^{-1}. \tag{9}$$

This allows the selected basis vectors to account for the impact of unselected vectors, yielding a more accurate approximation. Without this step, interactions among basis vectors are ignored, leading to suboptimal error reduction. Note that the refinement is applicable regardless of the parameter selection strategy. Figure 5 shows that refinement is crucial for reducing layer output error, presenting the layer output error after initialization with parameters selected by AdaAlloc. Further details on the error analysis are provided in Appendix C.3. Finally, $\boldsymbol{E}$ and $\boldsymbol{c}$ for channel $i$ are initialized to the selected indices and their refined values $\boldsymbol{x}^*$. Algorithm 1 summarizes the initialization process, with details provided in Appendix C.4.

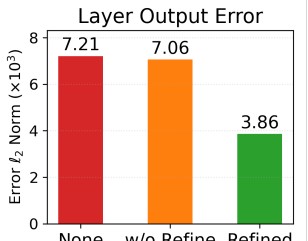

Figure 5: Effect of refinement on average layer output error.

## 4 EXPERIMENTS

We evaluate the effectiveness of QWHA in terms of model accuracy and training efficiency. We first compare QWHA with state-of-the-art QA-PEFT baseline and sparse high-rank adapters including FT-based adapters. Then, we provide a detailed analysis of the impact of using WHA and AdaAlloc. Finally, we demonstrate the efficiency of QWHA regarding WHA.

Table 3: Accuracy (%) evaluation results on CSQA and GSM8k benchmarks. 'QA Init.' denotes the existence of quantization-aware initialization.

| Bits | Method | Adapter Type | QA Init. | Coefficient Selection | LLaMA-3.1-8B | | LLaMA-3.2-3B | | Mistral-7B-v0.3 | |
|------|--------|--------------|----------|----------------------|------|-------|------|-------|------|-------|
| | | | | | CSQA | GSM8k | CSQA | GSM8k | CSQA | GSM8k |
| 16 | Pre-trained | - | - | - | 70.78 | 6.22 | 64.99 | 3.18 | 70.49 | 13.72 |
| | Fine-tuned | - | - | - | 71.84 | 59.74 | 66.43 | 44.80 | 71.87 | 54.51 |
| 4 | GPTQ$_{MagR}$ | - | - | - | 69.11 | 2.58 | 64.43 | 3.34 | 69.54 | 10.39 |
| | CLoQ | LoRA | ✓ | - | 69.58 | 53.83 | 65.48 | 39.27 | 71.32 | 52.01 |
| | SHiRA | Sparse | ✗ | Random | 71.07 | 54.36 | 63.10 | 40.71 | 70.88 | 51.02 |
| | LoCA | DCA | ✗ | LoCA | 71.45 | 54.36 | 65.59 | 40.33 | 71.55 | 47.99 |
| | SSH | DHA | ✗ | SSH | 70.75 | 53.98 | 65.83 | 39.80 | 71.57 | 47.99 |
| | **QWHA** | WHA | ✓ | AdaAlloc | **71.50** | **56.10** | **66.11** | **41.47** | **71.70** | **53.68** |
| 3 | GPTQ$_{MagR}$ | - | - | - | 67.76 | 2.65 | 61.49 | 2.43 | 67.57 | 1.29 |
| | CLoQ | LoRA | ✓ | - | 68.71 | 53.75 | 64.35 | 39.20 | 69.91 | 46.25 |
| | SHiRA | Sparse | ✗ | Random | 69.68 | 45.49 | 62.90 | 35.33 | 69.36 | 46.70 |
| | LoCA | DCA | ✗ | LoCA | 70.21 | 53.15 | 63.30 | 36.69 | 69.64 | 46.10 |
| | SSH | DHA | ✗ | SSH | 69.86 | 50.34 | 63.57 | 38.13 | 69.65 | 47.15 |
| | **QWHA** | WHA | ✓ | AdaAlloc | **70.50** | **55.34** | **64.80** | **39.58** | **70.22** | **47.84** |
| 2 | GPTQ$_{MagR}$ | - | - | - | 41.00 | 0.45 | 42.90 | 0.08 | 45.91 | 0.00 |
| | CLoQ | LoRA | ✓ | - | 56.49 | 33.89 | 54.89 | 26.53 | 61.80 | 33.36 |
| | SHiRA | Sparse | ✗ | Random | 51.84 | 27.74 | 52.91 | 22.59 | 59.08 | 33.57 |
| | LoCA | DCA | ✗ | LoCA | 56.71 | 33.97 | 53.87 | 23.88 | 62.03 | 33.89 |
| | SSH | DHA | ✗ | SSH | 56.06 | 30.55 | 54.01 | 25.77 | 62.31 | 32.06 |
| | **QWHA** | WHA | ✓ | AdaAlloc | **60.98** | **37.83** | **57.03** | **29.11** | **63.84** | **35.33** |

**Models and Datasets.** We evaluate QWHA on the Mistral-7B-v0.3 (Mistral AI, 2024) and LLaMA (Grattafiori et al., 2024) model families, including LLaMA-3.1-8B and LLaMA-3.2-3B. We evaluate the models on both general question-answering tasks for the models fine-tuned on instruction-following datasets and arithmetic reasoning tasks for the models fine-tuned on mathematical reasoning benchmarks. For instruction fine-tuning, we use the Stanford-Alpaca dataset (Taori et al., 2023)[1] with 52k samples. We evaluate on zero-shot commonsense question answering (CSQA)(Gao et al., 2024a), covering seven multiple-choice benchmarks(Clark et al., 2018; 2019; Zellers et al., 2019; Talmor et al., 2019; Bisk et al., 2020; Sakaguchi et al., 2021). For arithmetic reasoning, we fine-tune on the GSM8k (Cobbe et al., 2021) dataset and evaluate with zero-shot chain-of-thought reasoning questions on its test set, following Cobbe et al. (2021).

**Baselines.** We include full fine-tuned model (Fine-tuned) and quantized model, which use GPTQ (Frantar et al., 2023) with MagR (Zhang et al., 2024) (GPTQ$_{MagR}$) as baselines. We note that our method is also compatible with any other quantization schemes. We also include CLoQ, a recent QA-PEFT method that shares our goal of layer output error reduction during initialization for low-rank adapters. Other LoRA-based methods (Kim et al., 2024a; Liao et al., 2024) involving layer-wise calibration or layer-wise parameter allocation are orthogonal to our approach and can be integrated in future work. We evaluate sparse adapters, including SSH and LoCA (FT-based) and SHiRA (non FT-based). We note that LoCA further fine-tunes the randomly selected parameter indices via reparameterization with a cost of additional training overhead. We also build advanced hybrid baselines that integrate transforms or parameter selection strategies from prior works into our schemes by applying DCA and DHA with our AdaAlloc, or applying various parameter selection strategies to our WHA.

**Implementation Details.** Following prior work, adapters are applied to linear layers with a parameter budget of $P(r = 64)$, and quantization is performed with a group size of 64. Note that we apply a scaling factor $\alpha \simeq 1$ to all adapters, while the equations in the preceding sections omitted it by $\alpha = 1$ for simplicity. We set the AdaAlloc temperature to $t = 1$, which suffices to meet the full-rank condition. Further description on the training hyperparameter including scaling factor $\alpha$ and temperature $t$ are provided in Appendix D.1. We use WikiText-2 (Merity et al., 2016) as a

---

[1]https://huggingface.co/datasets/yahma/alpaca-cleaned

Table 4: Accuracy (%) evaluation results on CSQA and GSM8k benchmarks with variants of adapter types and parameter selection strategies in LLaMA-3.2-3B. 'QA Init.' denotes the existence of quantization-aware initialization, and 'Refine.' denotes the value refinement during initialization.

| Adapter Type | QA Init. | Coefficient Selection | Refine. | 4-bit | | 3-bit | | 2-bit | |
|---|---|---|---|---|---|---|---|---|---|
| | | | | CSQA | GSM8k | CSQA | GSM8k | CSQA | GSM8k |
| WHA | ✗ | Random | ✗ | 66.00 | 40.94 | 63.53 | 37.60 | 54.03 | 24.41 |
| WHA | ✓ | Random | ✓ | 65.91 | 40.71 | 63.91 | 37.30 | 54.48 | 24.48 |
| WHA | ✓ | Magnitude | ✓ | 66.07 | 41.01 | 64.52 | 36.69 | 56.49 | 28.12 |
| WHA | ✓ | LoCA | ✓ | 65.75 | 40.94 | 63.73 | 36.92 | 53.93 | 21.15 |
| WHA | ✓ | SSH | ✓ | 65.96 | 40.78 | 62.92 | 36.92 | 54.20 | 27.14 |
| **WHA** | ✓ | **AdaAlloc** | ✓ | **66.11** | **41.47** | **64.80** | **39.58** | **57.03** | **29.11** |
| DCA | ✓ | AdaAlloc | ✓ | 65.54 | 39.72 | 64.77 | 37.30 | 55.95 | 27.29 |
| DHA | ✓ | AdaAlloc | ✓ | 65.92 | 40.84 | 64.35 | 38.89 | 56.05 | 27.52 |
| Sparse | ✓ | AdaAlloc | ✓ | 65.60 | 40.94 | 63.43 | 37.53 | 55.97 | 26.54 |

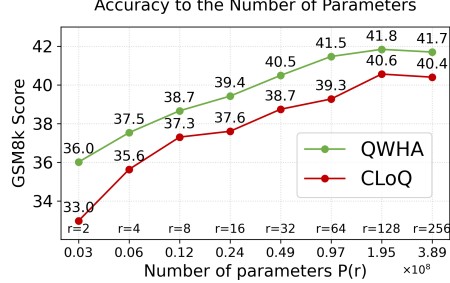

Figure 6: Accuracy of CLoQ and QWHA.

Table 5: Training time (hours) on Alpaca dataset.

| Batch Size | CLoQ | SHiRA | **QWHA** | SSH | LoCA |
|---|---|---|---|---|---|
| 1 | 12.5 | 15.5 | 18.2 | 63.3 | 92.3 |
| 2 | 7.1 | 8.2 | 9.7 | 45.8 | 53.4 |
| 4 | 5.0 | 5.5 | 6.0 | 26.1 | 30.1 |
| 8 | 4.1 | 4.3 | 4.6 | 13.3 | 16.5 |
| 16 | 3.6 | 3.7 | 3.9 | 8.3 | 9.8 |

calibration dataset for adapter initialization, following Deng et al. (2025), to ensure generality. All experiments are conducted on NVIDIA A100 80GB GPUs.

## 4.1 FINE-TUNED MODEL ACCURACY

**Main evaluation.** Table 3 shows that QWHA outperforms both low-rank adapters with quantization-aware initialization and conventional sparse adapters. In particular, the effectiveness of QWHA is evident in the 2-bit setting, where it achieves scores at least 2-3% higher than the baselines. Without quantization-aware initialization, sparse adapters, including FT-based adapters, perform worse than low-rank adapters in several cases. This underscores the need for quantization-aware initialization, especially in sub-4-bit settings where fine-tuning alone cannot fully restore performance. We note that task-specific results of the CSQA benchmark are presented in Appendix D.2.

**Effect of WHA and AdaAlloc.** We further examine the effectiveness of WHA and AdaAlloc, with QWHA consistently outperforming both low-rank adapters and advanced variants of sparse adapters. Figure 6 for 4-bit quantized LLaMA-3.2-3B shows that increasing the number of parameters in CLoQ cannot close the accuracy gap with QWHA, as QWHA with $P(r > 32)$ already surpasses CLoQ's maximum achievable score. This highlights the advantage of WHA, which provides superior representational capacity than low-rank adapters. Table 4 further demonstrates that WHA and AdaAlloc achieve the best results in each respective category of adapter type and parameter selection method. We note that LoCA's post-hoc location selection undermines the effectiveness of quantization-aware initialization based on the initially chosen parameters, unlike in PEFT. Ablations on the temperature $t$ in Equation 7 and quantization group size are provided in Appendix D.3 and D.4, respectively.

Table 6: Training time (hours) on Alpaca (Batch size 16).

| Method | CLoQ | SHiRA | SSH | LoCA | QWHA |
|---|---|---|---|---|---|
| Training time (hr) | 3.6 | 3.7 | 8.3 | 9.8 | 3.9 |

## 4.2 COMPUTATIONAL EFFICIENCY

Training time for QWHA on the Alpaca dataset with LLaMA-3.1-8B is reported in Table 5, where QWHA achieves a substantial speedup over previous FT-based adapters by leveraging WHA. WHA employs a single transform instead of the double transform used in conventional FT-based adapters, while achieving higher accuracy. Moreover, the fast recursive WHT kernel replaces matrix multiplications with a smaller number of additions and subtractions. In contrast, the recursive kernels of DCT and DHT, which require the DFT, are slower than direct matrix multiplication due to duplicated computations for imaginary parts. As a result, WHT achieves a similar training time to low-rank adapters or to a simple sparse adapter. In contrast, LoCA incurs additional latency even compared to SSH, due to the training of location parameters. Memory usage remains almost identical across all adapters with the same number of parameters. Detailed results on the training time of each transform kernel and the memory usage of each adapter are provided in Appendix D.5. In addition, initialization latency and memory usage of each adapters are provided in Appendix D.6, and inference throughput and memory usage are provided in Appendix D.7.

## 5 CONCLUSION

In this work, we introduce QWHA, a novel QA-PEFT framework featuring a Walsh-Hadamard transform-based adapter and its quantization-aware parameter initialization scheme. WHA offers strong fine-tuning capability and excels in quantization error reduction. The proposed AdaAlloc scheme facilitates both fine-tuning and quantization error reduction during parameter selection, while parameter refinement enables substantial quantization error reduction. We validate QWHA across diverse models and datasets, where it consistently outperforms existing baselines in accuracy and demonstrates its effectiveness. We also show that using WHA with a single transform provides computational benefits, enabling more efficient training than conventional FT-based adapters.

ACKNOWLEDGMENT

This work was supported in part by Institute of Information & communications Technology Planning & Evaluation (IITP) grant funded by the Korea government (MSIT) (No.RS-2025-02273157: Development of Low Power Training/Inference Technologies based on AI Semiconductors, RS-2023-00256081: artificial intelligence semiconductor support program to nurture the best talents, No. 2021-0-01343: Artificial Intelligence Graduate School Program (Seoul National University)), Samsung Research Funding Center under Project SRFC-TC1603-53, and BK21 FOUR program. (Corresponding Author: Jae-Joon Kim).

ETHICS STATEMENT

This paper presents research aimed at advancing the efficiency of large language models through quantization-aware parameter efficient fine-tuning. We acknowledge that large language models carry potential societal risks, including biases and misuse; however, our study focuses solely on methodological improvements in training and inference efficiency. We believe that our contributions do not introduce additional harms beyond those already inherent in the underlying models.

Regarding the use of large language models in this paper, we employed them as auxiliary tools for revising writing, checking grammar, and correcting typographical errors, but they did not play a significant role in research ideation or substantive writing that would warrant their consideration as contributors.

REPRODUCIBILITY STATEMENT

This paper introduces a novel algorithm for quantization-aware parameter efficient fine-tuning adapter design and its initialization. To ensure reproducibility, we provide the full source code as an anonymous, downloadable package in the supplementary materials, including detailed instructions for environmental setup, training, and evaluation. All datasets used in our experiments for fine-tuning and evaluation are publicly available, and we supply preprocessing steps and data preparation scripts to guarantee consistency with our reported results. Furthermore, complete proofs and derivations of our theoretical claims are presented in the appendix, with additional clarifications provided in the supplementary materials. Together, these resources are intended to fully enable independent reproduction and verification of our results.

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

## A  QUANTIZATION ERROR DISTRIBUTION

We present the distribution of quantization errors and their relationship to outliers in the pre-trained weights, as discussed in Section 2.1, in Figure 7. Figure 7(a) shows the overall error distribution, while Figure 7(b) highlights the channel-wise similarity between quantization errors and pre-trained weights in the 14th layer of LLaMA-3.2-3B. During quantization, values are divided by the quantization scale, typically defined per group within each output channel, and then rounded to an integer and clamped within a range determined by the bit-width. Most quantization errors remain within this rounding range, but large-magnitude outliers are often clamped, leading to large errors. Because model accuracy is highly sensitive to outlier weights, their quantization errors can significantly degrade performance. In QA-PEFT, it is therefore crucial to mitigate such outlier-induced errors during initialization by adapting the weights, particularly for large-magnitude values originating from salient outliers.

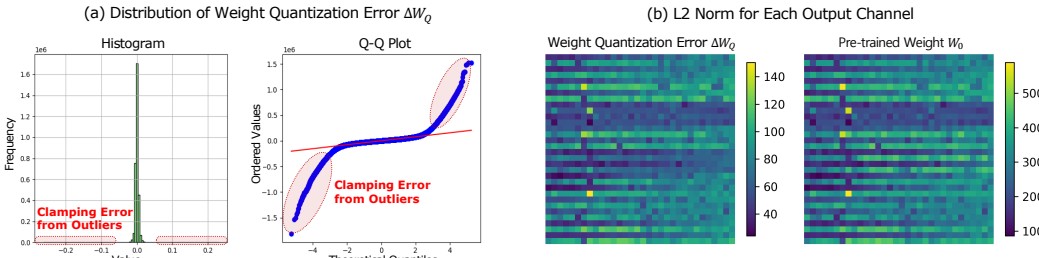

Figure 7: (a) Weight quantization error distribution and (b) its channel-wise similarity to the pre-trained weights in 14th layer Key projection of 4-bit quantized LLaMA-3.2-3B. In Figure (b), each pixel represents the $\ell_2$ norm of weight quantization errors (left) and that of pre-trained weights (right) for each output channel ordered by channel index from top-left to bottom-right.

## B  WHT-BASED ADAPTER (WHA)

### B.1  FT-BASED ADAPTER KERNELS

We describe a class of Fourier-related transform (FT) kernels employed in our adapters and prior studies in this section (Gao et al., 2024b; Du et al., 2025; Shen et al., 2025).

**Walsh-Hadamard Transform (WHT).**  The Walsh-Hadamard Transform (WHT) matrix $\boldsymbol{H}$ introduced in Equation 4 is constructed following conventions in prior works (Tseng et al., 2024; Ashkboos et al., 2024). For a dimension $N = 2^n$, a WHT matrix $\boldsymbol{H} \in \mathbb{R}^{N \times N}$ is defined recursively as the Kronecker product of smaller matrices:

$$\boldsymbol{H}_2 = \frac{1}{\sqrt{2}} \begin{bmatrix} 1 & 1 \\ 1 & -1 \end{bmatrix}, \quad \boldsymbol{H}_N = \boldsymbol{H}_2 \otimes \boldsymbol{H}_{2^{n-1}}, \tag{10}$$

where $\otimes$ denotes the Kronecker product. For non-power-of-two dimensions, Hadamard matrices exist for certain values (Seberry & Yamada, 1992; Hedayat et al., 1999; Gerakoulis & Ghassemzadeh, 2004), which can be retrieved from Sloane (2004). More generally, for $N = 2^n \cdot m$, where $\boldsymbol{H}_m$ is a known Hadamard matrix, the transform is defined as:

$$\boldsymbol{H}_N = \boldsymbol{H}_{2^n} \otimes \boldsymbol{H}_m. \tag{11}$$

The rows of $\boldsymbol{H}_N$ form an orthogonal basis, known as Walsh-Hadamard bases, satisfying:

$$\boldsymbol{H}_N^\top \boldsymbol{H}_N = \boldsymbol{H}_N \boldsymbol{H}_N^\top = \boldsymbol{I}_N. \tag{12}$$

The matrix $\boldsymbol{H}_{2^n}$ can be computed in $O(n \log n)$ time (Kunz, 1979). In practice, $\boldsymbol{H}_N$ can be precomputed once and cached for reuse across layers of the same size, incurring negligible cost in both computation and memory. To further accelerate computation, we employ the Fast Hadamard multiplication kernel from Dao-AILab (2024), which avoids explicit matrix construction by using a fused kernel of only additions and subtractions.

**Discrete Fourier Transform (DFT).**  FourierFT (Gao et al., 2024b) was the first study of FT-based adapters and used the discrete Fourier transform (DFT). The transform kernel $\boldsymbol{H} \in \mathbb{C}^{N \times N}$ is defined as:

$$H_{jk} = \frac{1}{\sqrt{N}} e^{-i \frac{2\pi jk}{N}} = \frac{1}{\sqrt{N}} \left\{ \cos\left(\frac{2\pi jk}{N}\right) - i \sin\left(\frac{2\pi jk}{N}\right) \right\}, \quad 0 \le j, k < N. \tag{13}$$

Although effective, later works adopted real-valued FT variants to avoid the complex-domain nature of the DFT, since deep learning frameworks typically discard the imaginary components and compute only with the real values.

**Discrete Hartley Transform (DHT).**  SSH (Shen et al., 2025) employs the discrete Hartley transform (DHT), a real-valued variant of the FT with kernel:

$$H_{jk} = \Re\left(\frac{1}{\sqrt{N}} e^{-i \frac{2\pi jk}{N}}\right) - \Im\left(\frac{1}{\sqrt{N}} e^{-i \frac{2\pi jk}{N}}\right) = \frac{1}{\sqrt{N}} \operatorname{cas}\left(\frac{2\pi jk}{N}\right), \quad 0 \le j, k < N, \tag{14}$$

where $\operatorname{cas}(x) = \cos x + \sin x$.

**Discrete Cosine Transform (DCT).**  LoCA (Du et al., 2025) employs another real-valued FT, the discrete cosine transform (DCT), whose kernel is:

$$H_{jk} = \begin{cases} \frac{1}{\sqrt{N}} & j = 0 \\ \sqrt{\frac{2}{N}} \cos\left(\frac{\pi(2k+1)j}{2N}\right) & 0 < j < N \end{cases}, \quad 0 \le k < N. \tag{15}$$

## B.2 Rank of WHA

This section provides a detailed explanation of the full-rank property of WHA and its conditions, as discussed in Section 3.1 and illustrated in Figure 2(a). To preserve the expressiveness of a fine-tuned model under a limited parameter budget, it is critical to ensure high rank capacity in the weight update. Unlike low-rank adapters, which inherently restrict the parameter subspace, WHA is sparsely structured yet can retain high representational capacity by maintaining full rank. This also holds in typical sparse adapters, including FT-based adapters.

We build on theoretical insights from prior work on sparse random matrices Coja-Oghlan et al. (2020), which provides conditions under which such matrices are full rank. Specifically, consider a random sparse matrix $\boldsymbol{F} \in \mathbb{R}^{d_{\text{out}} \times d_{\text{in}}}$, where each input and output channel has $k$ and $l$ non-zero entries on average. Then, $\boldsymbol{F}$ is full rank when $k, l \geq 2$ as $d_{\text{in}}, d_{\text{out}} \to \infty$, and thus full rank with high probability. Following the notations in Coja-Oghlan et al. (2020), we derive the corresponding condition for our setting to guarantee full-rank behavior in WHA.

**Condition Function.** We define the probability generating functions for the distributions of random non-zero entries per column and per channel. Given that these distributions are degenerate, the generating functions and their derivatives are:

$$D(z) = z^k, \quad D'(z) = kz^{k-1}, \quad D'(1) = k, \tag{16}$$

$$K(z) = z^l, \quad K'(z) = lz^{l-1}, \quad K'(1) = l, \tag{17}$$

Then, the condition function $\Phi(z)$ that determines the full rank condition is given by:

$$\Phi(z) = D\left(1 - \frac{K'(z)}{l}\right) - \frac{k}{l}\left[1 - K(z) - (1-z)K'(z)\right]. \tag{18}$$

To ensure the full rank of the matrix $A$, the inequality must hold as:

$$\Phi(z) < \Phi(0), \quad \forall 0 < z \leq 1, \tag{19}$$

Substituting the explicit forms for $D(z), K(z), D'(z), K'(z)$ into Equation B.2 yields the right hand side as:

$$\Phi(z) = (1 - z^{l-1})^k - \frac{k}{l} + kz^{l-1} - \frac{k(l-1)}{l}z^l. \tag{20}$$

As $\Phi(z = 0) = 1 - \frac{k}{l}$, the condition in Equation B.2 finally simplifies to:

$$(1 - z^{l-1})^k + kz^{l-1} - \frac{k(l-1)}{l}z^l - 1 < 0, \quad 0 < z \leq 1. \tag{21}$$

**Practical Considerations.** The inequality in Equation B.2 shows that the condition generally holds for integers $k, l \geq 2$. For the total number of parameters $p = r(d_{\text{in}} + d_{\text{out}})$ with $r \geq 2$, we have $k = p/d_{\text{in}} > r$ and $l = p/d_{\text{out}} > r$ under random selection, thus satisfying the full-rank condition when $d_{\text{in}}, d_{\text{out}}$ are sufficiently large. Importantly, AdaAlloc's per-channel allocation with remainder assignment and temperature control guarantees at least two elements in every channel (i.e., $l \geq 2$), which meets the sufficient condition required for the full-rank property. In addition, although AdaAlloc selects coefficient indices within each channel based on the magnitude of $\Delta W_Q H$, the coefficient distribution of $\Delta W_Q H$ under the WHT is close to a random normal distribution except for a small portion of outliers, as the correlations across input rows are nearly zero. Consequently, the selected index locations across input rows effectively behave like random choices. Empirically, parameter budgets corresponding to $P(r \geq 4)$ ensure at least two elements per row (i.e., $k \geq 2$), even for linear layers with large output dimensions, which might otherwise receive few parameters per input row. Hence, the full-rank conditions hold, and the matrix $\boldsymbol{F}$ in QWHA is nearly full rank.

### B.3 Energy Concentration of WHT

In this section, we quantify the energy concentration property of WHT discussed in Section 3.1, using Figure 2(b) and Figure 3(a).

**Distribution of Singular Values and Coefficients.** Figure 8 presents the distributions of singular values from SVD and of transform coefficients sorted by their squared magnitudes. Here, the area under each plot is equal to $\|\Delta W_Q\|_F^2$ (details in the following paragraph). The distributions follow a Pareto-like behavior, where sharpness can be quantified using the hill index $\eta$. The Pareto hill index is a value which implies the heaviness of a tail. In fact, this is the reciprocal of the Pareto tail index, defined as the mean of the log ratios of consecutive order statistics from the top-k largest magnitudes. A smaller hill index $\eta$ implies faster convergence of the cumulative distribution (Arnold, 1983). Hence, WHT exhibits the sharpest distribution, making it feasible to retain more information with fewer parameters $P(r)$ when implemented in a sparse adapter, as shown in Figure 2.

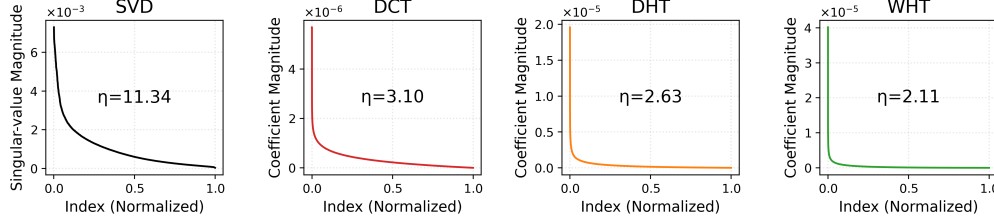

Singular Value and Coefficient Magnitude Distribution of $\Delta W_Q$ in each Decomposition and Transforms

Figure 8: Singular value and coefficient magnitude (squared) distributions with the Pareto hill index $\eta$ in the 14th-layer Key projection of LLaMA-3.2-3B.

**Energy of Singular Values and Coefficients.** Throughout this work, we use the term *energy* to denote the squared $\ell_2$ norm of the singular values in a decomposition or the spectral coefficients in a transform. We show that the total energy of both SVD and orthonormal transforms reduces to the Frobenius norm of the transformed matrix. (For example, in Figure 8, the area under each curve corresponds to $\|\Delta W_Q\|_F^2$.)

**Proposition 1.** Let $\boldsymbol{M} \in \mathbb{R}^{m \times n}$ be a matrix, and let its singular value decomposition (SVD) be $\boldsymbol{M} = \boldsymbol{U}\boldsymbol{\Sigma}\boldsymbol{V}^{\top}$, where $\boldsymbol{U} \in \mathbb{R}^{m \times m}$ and $\boldsymbol{V} \in \mathbb{R}^{n \times n}$ are orthonormal matrices, and $\boldsymbol{\Sigma} \in \mathbb{R}^{m \times n}$ is a diagonal matrix with entries $\sigma_i$ on the diagonal for $i = 1, \ldots, \min(m, n)$. Define $\boldsymbol{F} = \boldsymbol{M}\boldsymbol{H}$ for an orthonormal matrix $\boldsymbol{H} \in \mathbb{R}^{n \times n}$. Then, we have:

$$\|\boldsymbol{M}\|_F^2 = \sum_{i=1}^{\min(m,n)} \sigma_i^2 = \|\boldsymbol{F}\|_F^2.$$

**Proof.**

(i) *Identity* $\|\boldsymbol{M}\|_F^2 = \sum_{i=1}^{\min(m,n)} \sigma_i^2$.

Given $\boldsymbol{M} = \boldsymbol{U}\boldsymbol{\Sigma}\boldsymbol{V}^{\top}$, due to the orthonormality of $\boldsymbol{U}$, $\boldsymbol{M}^{\top}\boldsymbol{M}$ reduces to:

$$\boldsymbol{M}^{\top}\boldsymbol{M} = (\boldsymbol{U}\boldsymbol{\Sigma}\boldsymbol{V}^{\top})^{\top}(\boldsymbol{U}\boldsymbol{\Sigma}\boldsymbol{V}^{\top}) = \boldsymbol{V}\boldsymbol{\Sigma}^{\top}\boldsymbol{U}^{\top}\boldsymbol{U}\boldsymbol{\Sigma}\boldsymbol{V}^{\top} = \boldsymbol{V}\boldsymbol{\Sigma}^{\top}\boldsymbol{\Sigma}\boldsymbol{V}^{\top}. \tag{22}$$

Therefore, by the cyclic property of trace and the orthonormality of $\boldsymbol{V}$, $\|\boldsymbol{M}\|_F^2$ reduces to:

$$\|\boldsymbol{M}\|_F^2 = \operatorname{tr}(\boldsymbol{M}^{\top}\boldsymbol{M}) = \operatorname{tr}(\boldsymbol{V}\boldsymbol{\Sigma}^{\top}\boldsymbol{\Sigma}\boldsymbol{V}^{\top}) = \operatorname{tr}\left(\boldsymbol{\Sigma}^{\top}\boldsymbol{\Sigma}(\boldsymbol{V}^{\top}\boldsymbol{V})\right) = \operatorname{tr}(\boldsymbol{\Sigma}^{\top}\boldsymbol{\Sigma}) = \|\boldsymbol{\Sigma}\|_F^2. \tag{23}$$

Since $\boldsymbol{\Sigma}^{\top}\boldsymbol{\Sigma}$ is diagonal with entries $\sigma_i^2$ for $i = 1, \ldots, \min(m, n)$:

$$\|\boldsymbol{M}\|_F^2 = \|\boldsymbol{\Sigma}\|_F^2 = \sum_{i=1}^{\min(m,n)} \sigma_i^2. \tag{24}$$

(ii) *Identity* $\|\boldsymbol{M}\|_F^2 = \|\boldsymbol{F}\|_F^2$.

Given $\boldsymbol{F} = \boldsymbol{M}\boldsymbol{H}$, due to the orthonormality of $\boldsymbol{H}$:

$$\boldsymbol{F}^{\top}\boldsymbol{F} = (\boldsymbol{M}\boldsymbol{H})^{\top}(\boldsymbol{M}\boldsymbol{H}) = \boldsymbol{H}^{\top}\boldsymbol{M}^{\top}\boldsymbol{M}\boldsymbol{H}. \tag{25}$$

By the cyclic property of trace, $\|\boldsymbol{F}\|_F^2$ reduces to:

$$\|\boldsymbol{F}\|_F^2 = \operatorname{tr}\left((\boldsymbol{M}^{\top}\boldsymbol{M})(\boldsymbol{H}^{\top}\boldsymbol{H})\right) = \operatorname{tr}(\boldsymbol{M}^{\top}\boldsymbol{M}) = \|\boldsymbol{M}\|_F^2. \tag{26}$$

We note that this equivalence also applies to the coefficients defined with $\boldsymbol{F}' = \boldsymbol{H}'\Delta\boldsymbol{W}\boldsymbol{H}$, with $\boldsymbol{H}' \in \mathbb{R}^{m \times m}$, such that $\|\boldsymbol{M}\|_F^2 = \|\boldsymbol{F}\|_F^2 = \|\boldsymbol{F}'\|_F^2$.

**Outlier Reconstruction Ability of WHT.** Due to its energy concentration ability discussed above, WHT can reconstruct quantization errors with large outliers during initialization, which is critical for final model performance. Table 7 presents the numerical values corresponding to Figure 3(a), showing the proportion of outlier coefficients captured by each adapter.

Table 7: Percentage of outlier coefficients captured by each adapter under a parameter budget of $P(r = 64)$ in the $14^{\text{th}}$-layer Key projection of LLaMA-3.2-3B. Higher is better.

| Adapter Type | Query | Key | Value | Out | Gate | Up | Down | **Average** |
|---|---|---|---|---|---|---|---|---|
| DCA | 6.62 | 12.50 | 11.68 | 6.31 | 4.62 | 4.34 | 4.53 | 7.23 |
| DHA | 18.82 | 32.29 | 21.98 | 13.19 | 14.14 | 8.00 | 11.01 | 17.06 |
| **WHA** | 20.49 | 33.60 | 23.30 | 14.00 | 15.20 | 8.72 | 11.53 | **18.12** |

## C  QUANTIZATION-AWARE INITIALIZATION OF WHA

### C.1  OBJECTIVE FUNCTION

We reduce the original optimization objective in Equation 5 to the form in Equation 6, following the approach of Frantar et al. (2023); Deng et al. (2025). Using the notations from Section 3.2, the reduction proceeds as follows:

$$\|\Delta \boldsymbol{W}_Q \boldsymbol{X} - \boldsymbol{F}\boldsymbol{H}^{-1}\boldsymbol{X}\|_F^2 = \|(\Delta \boldsymbol{W}_Q - \boldsymbol{F}\boldsymbol{H}^{-1})\boldsymbol{X}\|_F^2 \tag{27}$$

$$= \operatorname{tr}\big((\Delta \boldsymbol{W}_Q - \boldsymbol{F}\boldsymbol{H}^{-1})\boldsymbol{X}\boldsymbol{X}^\top(\Delta \boldsymbol{W}_Q - \boldsymbol{F}\boldsymbol{H}^{-1})^\top\big) \tag{28}$$

$$= \operatorname{tr}\big((\Delta \boldsymbol{W}_Q - \boldsymbol{F}\boldsymbol{H}^{-1})\boldsymbol{R}\boldsymbol{R}^\top(\Delta \boldsymbol{W}_Q - \boldsymbol{F}\boldsymbol{H}^{-1})^\top\big) \tag{29}$$

$$= \|(\Delta \boldsymbol{W}_Q - \boldsymbol{F}\boldsymbol{H}^{-1})\boldsymbol{R}\|_F^2 \tag{30}$$

$$= \|\Delta \boldsymbol{W}_Q \boldsymbol{R} - \boldsymbol{F}\boldsymbol{H}^{-1}\boldsymbol{R}\|_F^2, \tag{31}$$

where $\boldsymbol{R} = \boldsymbol{U}\boldsymbol{\Sigma}^{1/2} \in \mathbb{R}^{d_{\text{in}} \times d_{\text{in}}}$ is an invertible square root of the Hessian Gram matrix $\boldsymbol{X}\boldsymbol{X}^\top$. This term is obtained by applying the SVD $\boldsymbol{X}\boldsymbol{X}^\top = \boldsymbol{U}\boldsymbol{\Sigma}\boldsymbol{U}^\top$, where $\boldsymbol{\Sigma}$ contains the eigenvalues on the diagonal and $\boldsymbol{U}$ is the matrix of orthonormal eigenvectors. Following Deng et al. (2025), we add a small regularization term $\lambda = 0.0001 \cdot \operatorname{tr}(\boldsymbol{X}\boldsymbol{X}^\top)/d_{\text{in}}$ to the diagonal if $\boldsymbol{R}$ is not originally invertible. This reduction allows us to replace $\boldsymbol{X}$ with $\boldsymbol{R}$, enabling efficient and effective calibration using multiple input data points. Rather than solving the optimization problem separately for each sample $\boldsymbol{X}$, we can accumulate the contribution of activations via $\boldsymbol{R}$ and solve a single reduced problem.

### C.2  PARAMETER SELECTION STRATEGIES

We present the parameter selection patterns of each method discussed in Section 3.2. As shown in Figure 9, magnitude-based selection allocates parameters to a limited number of channels, while conventional methods such as SSH and LoCA incorporate random selection to avoid rank reduction. However, these approaches fail to reduce quantization error during initialization because the selected parameters are not optimal for error reconstruction. In contrast, AdaAlloc identifies the most important locations within each channel while preventing rank reduction through per-channel budgets, thereby providing the most effective initialization and fine-tuning.

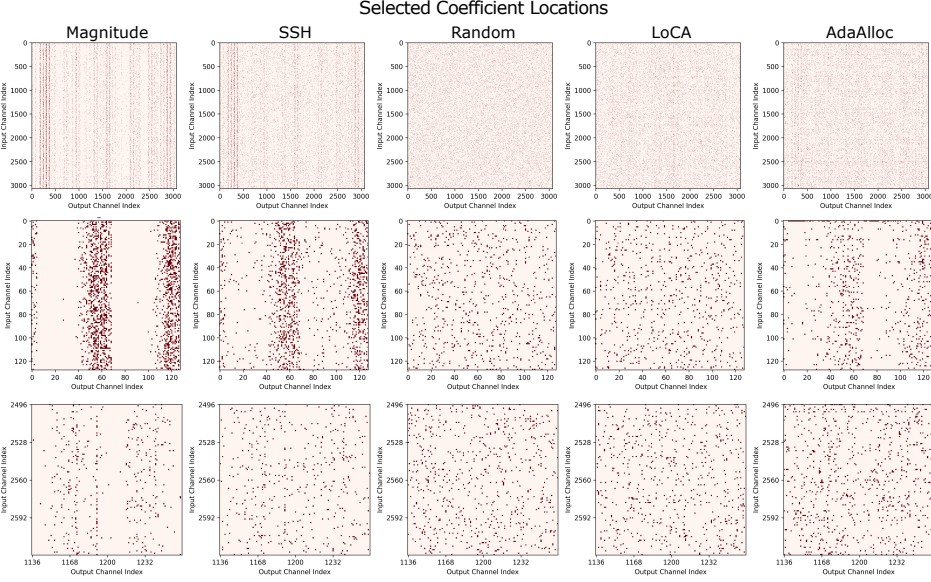

Figure 9: Parameter selection patterns and two example zoomed-in results of each method in the 14[th]-layer Query projection of LLaMA-3.2-3B.

## C.3 VALUE REFINEMENT

We present the layer output error after WHA initialization with and without value refinement in Table 8, as discussed in Figure 5 in Section 3.2. Without refinement of the selected coefficients in the initial dense solution matrix $\Delta W_Q H$, correlations among the columns are ignored, and the impact of sparsifying other columns cannot be considered, leading to suboptimal error reconstruction.

Table 8: Layer output error ($\ell_2$ norm, scaled by $\times 10^3$) after initialization with and without value refinement in 4-bit quantized LLaMA-3.2-3B. Parameters are selected by AdaAlloc. 'None' denotes the error before initialization.

| Method | Query | Key | Value | Out | Gate | Up | Down | Average |
|--------|-------|-----|-------|-----|------|-----|------|---------|
| None | 13.84 | 0.54 | 28.08 | 4.66 | 1.88 | 25.76 | 21.36 | 7.21 |
| W/o Refinement | 11.39 | 0.62 | 26.21 | 4.39 | 2.11 | 24.33 | 20.97 | 7.06 |
| **Refined** | **5.11** | **0.27** | **14.92** | **2.01** | **1.13** | **17.97** | **15.25** | **3.86** |

## C.4 CHANNEL-WISE PARAMETER SELECTION AND INITIALIZATION

We provide a detailed description of the formulation and solution of the sparse approximation problem underlying Algorithm 1, based on the notations in Section 3.2.

**Sparse Approximation Problem.** With a channel-wise breakdown of the objective in Equation 6, the goal is to initialize the $i$-th channel of the parameter matrix $F$, denoted $F_{i,:}$, given the per-channel parameter budget $p_i$. The objective is for $F_{i,:} H^{-1} R$ to closely approximate the projected quantization error $(\Delta W_Q)_{i,:} R$ in the $\ell_2$ sense. As we constrain $F_{i,:}$ to have exactly $p_i$ non-zero elements, the term $F_{i,:} H^{-1} R$ becomes a sparse linear combination of standard basis vectors:

$$F_{i,:} H^{-1} R = \sum_{k=0}^{p_i} F_{i,j_k} e^{(j_k)} H^{-1} R, \tag{32}$$

where $e^{(j_k)}$ is the $j_k$-th standard basis vector. Since $e^{(j_k)} H^{-1} R$ corresponds to the $j_k$-th channel of $H^{-1} R$, the problem reduces to selecting $p_i$ rows from $H^{-1} R$ that best approximate $(\Delta W_Q)_{i,:} R$.

**Greedy Algorithm for Sparse Approximation.** The problem generalizes to a standard sparse approximation problem: given a full set of basis vectors $\beta = \{u_1, u_2, \ldots, u_n\}$ with each $u_i \in \mathbb{R}^d$, we aim to select $k$ vectors whose linear combination best approximates a target vector $v \in \mathbb{R}^d$. We represent the sparse coefficient vector as $x = [x_1, x_2, \ldots, x_k] \in \mathbb{R}^k$, corresponding to the selected $k$ basis vectors. Formally, we solve:

$$\min_x \|v - xB\|_2^2 \quad \text{subject to} \quad \|x\|_0 = k, \tag{33}$$

where $B \in \mathbb{R}^{k \times d}$ is a submatrix formed from selected rows of the original basis. In our setting, $B = H^{-1} R$, $v = (\Delta W_Q)_{i,:} R$, and $k = p_i$.

Since this problem is NP-hard, we adopt a greedy approximation. We first compute $x = vB^{-1} = \Delta W_Q H$, which is in fact the non-sparse solution to the objective in Equation 6, and select the $k$ entries of $x$ with the largest magnitudes. Let the corresponding indices be $i_1, i_2, \ldots, i_k$, and define the selected basis $B' = [u_{i_1}; \ldots; u_{i_k}]$. We then solve a least-squares problem over the selected support:

$$x^* = \operatorname{argmin}_{(x_{i_1}, \cdots, x_{i_k})} \|[x_{i_1}\ x_{i_2}\ \cdots\ x_{i_k}]B' - v\|_2^2 = vB'^{\top}(B'B'^{\top})^{-1}. \tag{34}$$

While this solution is numerically optimal when $B$ is orthogonal, we empirically demonstrate its effectiveness under general conditions. Combined with our AdaAlloc-based parameter allocation strategy, this initialization consistently yields high quantization error reconstruction ability while maintaining full rank capacity.

# D EXPERIMENTAL DETAILS AND ABLATIVE STUDY

## D.1 FINE-TUNING HYPERPARAMETERS

We follow the hyperparameter settings adapted from Gao et al. (2024b) and Deng et al. (2025). Training is performed using the AdamW optimizer (Loshchilov & Hutter, 2017). Table 9 reports the key settings, including minibatch size, weight decay, dropout ratio, learning rate scheduler, maximum sequence length, number of training epochs, warmup ratio, and the adapter scaling factor $\alpha$, which can be applied to adapters in Equation 2 through Equation 4. Due to an implementation detail in our codebase, the explicitly specified scaling factor is internally divided by the layer input dimension $d_{\text{in}}$. As a consequence, the actual scaling applied during training is $\alpha_{\text{effective}} = \alpha_{\text{explicit}}/d_{\text{in}}$. To match the effective scaling used in CLoQ (i.e., $\alpha_{\text{effective}} \approx 1.0$), we set the explicit scaling factor to $\alpha_{\text{explicit}} = 4000$, which is close to the typical input dimensions of our models (3072 for LLaMA-3.2-3B and 4096 for LLaMA-3.1-8B and Mistral-7B-v0.3). Under this implementation, the effective scaling becomes $\alpha_{\text{effective}} = 4000/d_{\text{in}} \approx 1.0$, ensuring consistent gradient scaling between low-rank.

The learning rates for each combination of model, task, method, and bit-width are summarized in Table 10. We note that 128 sequences of length 2048, randomly sampled from the WikiText-2 (Merity et al., 2016) dataset, are used as a calibration set for quantization and adapter initialization, as these processes are robust to the choice of dataset (Frantar et al., 2023; Zhang et al., 2024; Deng et al., 2025). The total number of parameters for $P(r = 64)$ is reported in Table 11, broken down by each projection, the layers containing these projections, and the entire model.

Table 9: Hyperparameter settings for Alpaca and GSM8K training

| Dataset | Alpaca | | GSM8K | |
|---|---|---|---|---|
| **Method** | **CLoQ** | **QWHA** | **CLoQ** | **QWHA** |
| Optimizer | | AdamW | | |
| Batch Size | | 64 | | |
| LR Scheduler | | cosine | | |
| Max Sequence Length | | 512 | | |
| Epochs | | 3 | | 6 |
| Warmup Ratio | | 0.1 | | 0.03 |
| Weight Decay | | 1 | | 0.1 |
| Dropout | 0.1 | 0 | 0.1 | 0 |
| Scale | 1 | 4000 / $d_{in}$ | 1 | 4000 / $d_{in}$ |

Table 10: Learning rate for each model and bit widths on Alpaca and GSM8K training.

| Model | Bits | Alpaca | | GSM8K | |
|---|---|---|---|---|---|
| | | **CLoQ** | **QWHA** | **CLoQ** | **QWHA** |
| Llama-3.1-8B | 4 | 1e-5 | 3e-5 | 1e-4 | 5e-5 |
| | 3 | 1e-5 | 3e-5 | 1e-4 | 7e-5 |
| | 2 | 1e-5 | 2e-5 | 7e-5 | 5e-5 |
| Llama-3.2-3B | 4 | 1e-4 | 3e-5 | 1e-4 | 7e-5 |
| | 3 | 1e-4 | 3e-5 | 1e-4 | 1e-4 |
| | 2 | 2e-4 | 5e-5 | 1e-4 | 2e-4 |
| Mistral-7B-v0.3 | 4 | 3e-5 | 5e-6 | 3e-5 | 2e-5 |
| | 3 | 2e-5 | 5e-6 | 3e-5 | 3e-5 |
| | 2 | 2e-5 | 7e-6 | 3e-5 | 3e-5 |

Table 11: Total number of parameters at $P(r = 64)$ for each projection, layer, and model.

| Model | q_proj | k_proj | v_proj | o_proj | gate_proj | up_proj | down_proj | per-layer | per-model |
|---|---|---|---|---|---|---|---|---|---|
| LLaMA-3.1-8B | 524288 | 327680 | 327680 | 524288 | 1179648 | 1179648 | 1179648 | 5242880 | 167772160 |
| LLaMA-3.2-3B | 393216 | 262144 | 262144 | 393216 | 720896 | 720896 | 720896 | 3473408 | 97255424 |
| Mistral-7B-v0.3 | 524288 | 327680 | 327680 | 524288 | 1179648 | 1179648 | 1179648 | 5242880 | 167772160 |

## D.2 COMMONSENSE QUESTION-ANSWERING BENCHMARK

We present the detailed accuracy scores on the commonsense question answering (CSQA) benchmark in this section, while the averaged scores are reported in Section 4. The results for each of the seven individual tasks, are provided in Table 12 and Table 13, corresponding to Table 3 and Table 4, respectively.

Table 12: Accuracy (%) evaluation results of CSQA benchmarks.

| Model | Bits | Method | Arc-c | Arc-e | BoolQ | Hella. | Obqa | Piqa | Wino. | Average |
|---|---|---|---|---|---|---|---|---|---|---|
| LLaMA-3.1-8B | 16 | Pre-trained | 53.41 | 81.10 | 82.11 | 78.91 | 44.80 | 81.23 | 73.88 | 70.78 |
| | | Fine-tuned | 56.40 | 81.57 | 81.99 | 79.85 | 46.40 | 81.99 | 74.66 | 71.84 |
| | 4 | GPTQ$_{MagR}$ | 48.98 | 78.96 | 81.77 | 75.87 | 45.00 | 79.87 | 73.32 | 69.11 |
| | | CLoQ | 50.34 | 79.50 | 83.12 | 75.63 | 45.40 | 79.92 | 73.16 | 69.58 |
| | | SHiRA | 53.24 | 81.14 | 82.81 | 78.76 | 46.60 | 81.56 | 73.40 | 71.07 |
| | | LoCA | 55.12 | 80.98 | 83.06 | 79.56 | 47.60 | 81.56 | 72.30 | 71.45 |
| | | SSH | 54.69 | 78.62 | 83.18 | 79.44 | 44.60 | 81.56 | 73.16 | 70.75 |
| | | **QWHA** | 55.20 | 80.26 | 83.64 | 79.62 | 47.00 | 81.99 | 72.77 | **71.50** |
| | 3 | GPTQ$_{MagR}$ | 48.81 | 76.52 | 81.59 | 73.74 | 43.80 | 78.24 | 71.59 | 67.76 |
| | | CLoQ | 49.91 | 77.48 | 82.42 | 74.30 | 44.80 | 79.05 | 73.01 | 68.71 |
| | | SHiRA | 53.41 | 79.17 | 80.98 | 77.63 | 44.80 | 80.30 | 71.51 | 69.68 |
| | | LoCA | 54.61 | 79.88 | 81.13 | 78.25 | 45.20 | 80.25 | 72.14 | 70.21 |
| | | SSH | 53.24 | 75.97 | 82.81 | 80.27 | 43.00 | 81.56 | 72.14 | 69.86 |
| | | **QWHA** | 54.69 | 80.05 | 81.68 | 78.70 | 45.20 | 80.79 | 72.38 | **70.50** |
| | 2 | GPTQ$_{MagR}$ | 24.23 | 38.51 | 53.15 | 36.73 | 25.00 | 57.67 | 51.70 | 41.00 |
| | | CLoQ | 37.80 | 56.19 | 67.74 | 64.14 | 35.40 | 72.31 | 61.88 | 56.49 |
| | | SHiRA | 33.28 | 51.31 | 64.31 | 56.76 | 31.40 | 68.72 | 57.14 | 51.84 |
| | | LoCA | 37.63 | 59.22 | 68.13 | 63.79 | 35.60 | 70.95 | 61.64 | 56.71 |
| | | SSH | 39.08 | 57.79 | 67.58 | 62.00 | 35.00 | 71.38 | 59.59 | 56.06 |
| | | **QWHA** | 41.72 | 64.94 | 74.62 | 68.11 | 37.40 | 74.54 | 65.51 | **60.98** |
| LLaMA-3.2-3B | 16 | Pre-trained | 45.99 | 71.63 | 73.39 | 73.61 | 43.00 | 77.48 | 69.85 | 64.99 |
| | | Fine-tuned | 48.29 | 73.15 | 74.95 | 76.71 | 43.80 | 77.91 | 70.17 | 66.43 |
| | 4 | GPTQ$_{MagR}$ | 44.97 | 70.83 | 74.95 | 71.34 | 42.60 | 77.15 | 69.14 | 64.43 |
| | | CLoQ | 47.70 | 72.35 | 74.95 | 74.25 | 42.40 | 77.86 | 68.82 | 65.48 |
| | | SHiRA | 43.17 | 68.01 | 71.80 | 73.33 | 41.60 | 76.99 | 66.85 | 63.10 |
| | | LoCA | 47.78 | 73.40 | 74.34 | 74.33 | 42.20 | 78.13 | 68.98 | 65.59 |
| | | SSH | 47.95 | 73.32 | 75.35 | 74.38 | 42.80 | 78.67 | 68.35 | 65.83 |
| | | **QWHA** | 48.98 | 73.15 | 75.78 | 74.44 | 41.60 | 79.00 | 69.85 | **66.11** |
| | 3 | GPTQ$_{MagR}$ | 42.92 | 66.08 | 70.86 | 68.29 | 40.00 | 76.12 | 66.14 | 61.49 |
| | | CLoQ | 46.33 | 71.25 | 72.97 | 72.41 | 41.40 | 78.35 | 67.72 | 64.35 |
| | | SHiRA | 44.54 | 68.43 | 72.26 | 71.31 | 40.40 | 77.75 | 65.67 | 62.90 |
| | | LoCA | 44.71 | 69.74 | 70.83 | 72.17 | 41.20 | 78.13 | 66.30 | 63.30 |
| | | SSH | 44.88 | 70.37 | 71.62 | 72.17 | 41.80 | 78.18 | 65.98 | 63.57 |
| | | **QWHA** | 47.18 | 72.64 | 72.51 | 72.72 | 41.80 | 79.71 | 67.01 | **64.80** |
| | 2 | GPTQ$_{MagR}$ | 26.62 | 38.89 | 54.28 | 39.32 | 29.00 | 59.36 | 52.80 | 42.90 |
| | | CLoQ | 35.24 | 56.27 | 66.02 | 59.77 | 37.40 | 70.35 | 59.19 | 54.89 |
| | | SHiRA | 33.28 | 56.40 | 63.64 | 55.11 | 34.80 | 69.86 | 57.30 | 52.91 |
| | | LoCA | 34.64 | 58.42 | 64.46 | 56.26 | 34.60 | 71.44 | 57.30 | 53.87 |
| | | SSH | 34.81 | 58.33 | 64.65 | 56.21 | 36.00 | 70.51 | 57.54 | 54.01 |
| | | **QWHA** | 37.29 | 61.99 | 65.26 | 61.76 | 37.20 | 73.88 | 61.80 | **57.03** |
| Mistral-7B-v0.3 | 16 | Pre-trained | 52.30 | 78.24 | 82.14 | 80.42 | 44.20 | 82.26 | 73.88 | 70.49 |
| | | Fine-tuned | 55.03 | 80.05 | 84.19 | 81.09 | 45.80 | 82.43 | 74.51 | 71.87 |
| | 4 | GPTQ$_{MagR}$ | 51.37 | 76.52 | 80.55 | 79.71 | 44.00 | 81.72 | 72.93 | 69.54 |
| | | CLoQ | 54.52 | 78.58 | 83.91 | 81.09 | 44.00 | 82.37 | 74.74 | 71.32 |
| | | SHiRA | 53.50 | 78.41 | 82.69 | 80.46 | 44.60 | 82.43 | 74.11 | 70.88 |
| | | LoCA | 53.84 | 78.28 | 83.88 | 81.12 | 45.40 | 82.86 | 75.45 | 71.55 |
| | | SSH | 53.75 | 78.45 | 84.71 | 81.18 | 45.60 | 82.64 | 74.66 | 71.57 |
| | | **QWHA** | 54.69 | 78.79 | 84.74 | 80.93 | 45.40 | 82.70 | 74.66 | **71.70** |
| | 3 | GPTQ$_{MagR}$ | 49.23 | 75.00 | 77.06 | 78.22 | 42.20 | 80.63 | 70.64 | 67.57 |
| | | CLoQ | 52.99 | 77.44 | 80.55 | 80.37 | 43.40 | 81.39 | 73.24 | 69.91 |
| | | SHiRA | 51.37 | 77.31 | 79.57 | 79.52 | 44.20 | 81.45 | 72.14 | 69.36 |
| | | LoCA | 51.96 | 77.31 | 80.95 | 80.17 | 43.40 | 81.39 | 72.30 | 69.64 |
| | | SSH | 51.54 | 77.23 | 81.99 | 79.93 | 43.20 | 81.39 | 72.30 | 69.65 |
| | | **QWHA** | 52.56 | 76.94 | 82.32 | 80.31 | 44.20 | 82.21 | 73.01 | **70.22** |
| | 2 | GPTQ$_{MagR}$ | 26.37 | 41.46 | 54.37 | 48.88 | 29.80 | 62.62 | 57.85 | 45.91 |
| | | CLoQ | 43.94 | 66.46 | 74.25 | 70.95 | 38.00 | 75.41 | 63.61 | 61.80 |
| | | SHiRA | 39.33 | 64.27 | 71.38 | 66.58 | 36.60 | 73.88 | 61.56 | 59.08 |
| | | LoCA | 43.77 | 67.63 | 75.99 | 69.78 | 37.80 | 74.54 | 64.72 | 62.03 |
| | | SSH | 44.03 | 68.77 | 76.79 | 70.15 | 36.00 | 75.35 | 65.11 | 62.31 |
| | | **QWHA** | 45.39 | 69.78 | 78.53 | 71.97 | 37.60 | 76.44 | 67.17 | **63.84** |

Table 13: Accuracy (%) evaluation results of CSQA benchmarks on LLaMA-3.2-3B.

| Bits | Adapter Type | QA Init. | Coeff. Selection | Refine. | Arc-c | Arc-e | BoolQ | Hella. | OBQA | PiQA | Wino. | **Average** |
|---|---|---|---|---|---|---|---|---|---|---|---|---|
| 4 | WHA | ✗ | Random | ✗ | 48.55 | 73.70 | 74.53 | 74.43 | 43.20 | 78.67 | 68.90 | 66.00 |
| | WHA | ✓ | Random | ✓ | 48.04 | 73.36 | 74.22 | 74.50 | 42.00 | 78.78 | 70.48 | 65.91 |
| | WHA | ✓ | Magnitude | ✓ | 48.81 | 72.01 | 75.26 | 74.39 | 42.80 | 79.43 | 69.77 | 66.07 |
| | WHA | ✓ | LoCA | ✓ | 47.61 | 73.06 | 74.16 | 74.18 | 43.20 | 78.56 | 69.46 | 65.75 |
| | WHA | ✓ | SSH | ✓ | 47.61 | 73.40 | 75.26 | 74.45 | 43.00 | 78.07 | 69.93 | 65.96 |
| | WHA | ✓ | AdaAlloc | ✓ | 48.98 | 73.15 | 75.78 | 74.44 | 41.60 | 79.00 | 69.85 | **66.11** |
| | DCA | ✓ | AdaAlloc | ✓ | 47.10 | 72.47 | 73.24 | 75.14 | 43.00 | 79.11 | 68.75 | 65.54 |
| | DHA | ✓ | AdaAlloc | ✓ | 47.70 | 73.06 | 75.60 | 74.61 | 41.60 | 78.67 | 70.17 | 65.92 |
| | Sparse | ✓ | AdaAlloc | ✓ | 47.61 | 72.52 | 73.49 | 75.03 | 43.60 | 78.13 | 68.82 | 65.60 |
| 3 | WHA | ✗ | Random | ✗ | 44.88 | 70.54 | 71.53 | 71.71 | 41.40 | 77.86 | 66.77 | 63.53 |
| | WHA | ✓ | Random | ✓ | 44.97 | 71.38 | 71.38 | 72.38 | 41.80 | 78.62 | 66.85 | 63.91 |
| | WHA | ✓ | Magnitude | ✓ | 47.35 | 72.31 | 72.69 | 72.40 | 41.00 | 79.38 | 66.54 | 64.52 |
| | WHA | ✓ | LoCA | ✓ | 46.25 | 69.49 | 71.38 | 71.93 | 41.80 | 78.51 | 66.77 | 63.73 |
| | WHA | ✓ | SSH | ✓ | 44.45 | 67.38 | 71.83 | 71.38 | 40.20 | 78.35 | 66.85 | 62.92 |
| | WHA | ✓ | AdaAlloc | ✓ | 47.18 | 72.64 | 72.51 | 72.72 | 41.80 | 79.71 | 67.01 | **64.80** |
| | DCA | ✓ | AdaAlloc | ✓ | 46.08 | 72.05 | 73.55 | 73.11 | 42.20 | 78.18 | 68.19 | 64.77 |
| | DHA | ✓ | AdaAlloc | ✓ | 45.31 | 72.18 | 72.32 | 72.91 | 41.40 | 78.94 | 67.40 | 64.35 |
| | Sparse | ✓ | AdaAlloc | ✓ | 45.65 | 70.75 | 68.35 | 72.77 | 41.00 | 77.58 | 67.96 | 63.43 |
| 2 | WHA | ✗ | Random | ✗ | 34.56 | 59.05 | 65.17 | 55.68 | 34.80 | 70.67 | 58.25 | 54.03 |
| | WHA | ✓ | Random | ✓ | 34.90 | 58.54 | 64.31 | 58.13 | 35.40 | 70.84 | 59.27 | 54.48 |
| | WHA | ✓ | Magnitude | ✓ | 37.71 | 60.65 | 64.83 | 60.88 | 37.40 | 72.58 | 61.40 | 56.49 |
| | WHA | ✓ | LoCA | ✓ | 33.96 | 58.08 | 64.71 | 56.27 | 35.80 | 70.67 | 58.01 | 53.93 |
| | WHA | ✓ | SSH | ✓ | 34.39 | 55.77 | 64.50 | 58.80 | 36.00 | 70.73 | 59.19 | 54.20 |
| | WHA | ✓ | AdaAlloc | ✓ | 37.29 | 61.99 | 65.26 | 61.76 | 37.20 | 73.88 | 61.80 | **57.03** |
| | DCA | ✓ | AdaAlloc | ✓ | 35.92 | 57.15 | 66.18 | 61.94 | 37.00 | 72.58 | 60.85 | 55.95 |
| | DHA | ✓ | AdaAlloc | ✓ | 36.09 | 58.59 | 66.67 | 61.12 | 36.00 | 72.52 | 61.33 | 56.05 |
| | Sparse | ✓ | AdaAlloc | ✓ | 35.67 | 56.40 | 67.16 | 63.06 | 36.40 | 73.23 | 59.91 | 55.97 |

### D.3 ABLATION ON TEMPERATURE $t$

We evaluate the effect of the temperature parameter $t$ in Equation 7 on the fine-tuned accuracy of LLaMA-3.2-3B under 4-bit quantization using the parameter budget $P(r = 64)$. The results show that model accuracy remains stable for temperatures between 0.5 and 1.0, while performance slightly degrades when $t$ is too small or too large. A low temperature distributes the parameter budget nearly uniformly across channels, preventing sufficient allocation to channels with large quantization errors. Conversely, an overly high temperature over-concentrates parameters in these large-magnitude channels, neglecting important coefficients within low-magnitude channels and assigning unnecessary parameters within high-magnitude ones. As a result, both excessively low and excessively high temperatures lead to decreased fine-tuned model accuracy. Within the robust range of $t \in [0.5, 1.0]$, we selected $t = 1$ as the default setting, since it naturally favors allocating more parameters to outlier-included, large-magnitude channels while still maintaining stable empirical performance and methodological simplicity.

Table 14: Effect of temperature $t$ on GSM8k accuracy of LLaMA-3.2-3B under 4-bit quantization.

| Temperature $t$ | 0.25 | 0.5 | 1.0 | 1.5 | 2.0 |
|---|---|---|---|---|---|
| GSM8k Acc. (%) | 40.11 | 41.39 | 41.47 | 40.64 | 40.04 |

### D.4 ABLATION ON QUANTIZATION GROUP SIZE

We conduct an ablation study on the effect of quantization group size in the LLaMA-3.2-3B model using 2-bit quantization, where the impact of group size on model accuracy is most clearly observed, as shown in Table 15. Smaller group sizes provide finer granularity, leading to higher model accuracy. However, they also incur greater computational overhead during the quantization and dequantization process due to the increased number of quantization parameters. Considering this trade-off, we adopt a group size of 64 for our experiments, consistent with prior works on quantization-aware PEFT.

Table 15: GSM8k accuracy (%) of QWHA on LLaMA-3.2-3B with 2-bit quantization using various quantization group sizes.

| Group Size | 32 | 64 | 128 | 256 |
|---|---|---|---|---|
| Score | 29.94 | 29.11 | 24.48 | 22.51 |

### D.5 TRAINING OVERHEAD OF ADAPTERS

**Train time**   We compare the training time of adapters using single-transform and two-transform designs on both WHT and conventional transform kernels, such as the DCT used in LoCA and the DHT used in SSH, in Table 16. While employing WHA reduces training time, applying it with a single transform further decreases computation. The impact of a single transform is especially evident in DCT and DHT, where training time is substantially reduced since their computational overhead due to the transform is larger than that of WHT. Note that DCT and DHT have identical training times, as their computational cost is the same and differs only in the element values within the transform kernel. Our proposed WHA employs a 1D WHT in the context of quantization, whereas conventional FT-based PEFT methods such as LoCA and SSH use 2D DCT and 2D DHT, respectively.

Table 16: Training time (hours) of FT-based adapters with different transform kernels on LLaMA-3.1-8B with the Alpaca dataset.

| Batch Size | WHT | | DCT / DHT | |
|:---:|:---:|:---:|:---:|:---:|
| | 1D | 2D | 1D | 2D |
| 1 | 18.2 | 25.3 | 46.2 | 63.3 |
| 2 | 9.7 | 13.1 | 32.1 | 45.8 |
| 4 | 6.0 | 8.0 | 17.4 | 26.1 |
| 8 | 4.6 | 5.5 | 9.0 | 13.3 |
| 16 | 3.9 | 4.3 | 6.7 | 8.3 |

**Memory Usage**   We report the memory usage of each method under the same experimental setting as in Section 4, using NVIDIA A100 80GB GPU. As shown in Table 17, QWHA shows memory usage comparable to LoRA. SSH also exhibits similar memory usage as QWHA, since the only difference between them is the computation with a pre-defined transform kernel matrix. Since this matrix is shared across layers, the memory overhead is negligible. In contrast, LoCA incurs additional memory consumption due to the training of location parameters, resulting in a few gigabytes of overhead depending on the batch size.

Table 17: GPU memory usage (GB) during fine-tuning on LLaMA-3.1-8B with 4-bit quantization using the Alpaca dataset. All adapters use the same number of trainable parameters with $P(r = 64)$.

| Batch Size | CLoQ | QWHA | LoCA |
|:---|:---:|:---:|:---:|
| 1 | 22.1 | 22.1 | 23.3 |
| 2 | 26.6 | 27.2 | 28.4 |
| 4 | 32.6 | 33.2 | 34.4 |
| 8 | 44.7 | 45.3 | 46.4 |
| 16 | 68.8 | 69.4 | 70.6 |

## D.6 INITIALIZATION OVERHEAD OF ADAPTERS

**Initialization Time.** Table 18 reports the initialization latency of the low-rank adapter (CLoQ) and the FT-based adapters (QWHA and SSH) under the 4-bit setting across different models. The initialization of CLoQ requires gathering activations and quantization errors, followed by SVD decomposition before solving the least-squares problem. In contrast, the initialization of FT-based adapters involves collecting activations and quantization errors, followed by parameter selection and value assignment through channel-wise transforms and solving least-squares problem. During this process, the fast Hadamard kernel allows QWHA with WHT to perform efficient computation. As a result, QWHA achieves comparable initialization time to CLoQ, while SSH with the DHT kernel incurs significantly higher latency.

Table 18: Initialization latency (hours) of each method under the 4-bit setting.

| Method | LLaMA-3.2-3B | LLaMA-3.1-8B | Mistral-7B-v0.3 |
|--------|-------------|-------------|-----------------|
| CLoQ | 0.58 | 1.14 | 1.26 |
| SSH | 3.85 | 8.09 | 8.58 |
| QWHA | 0.66 | 1.34 | 1.46 |

**Memory Usage.** We measure the peak memory consumption during initialization for each method on the LLaMA-3.2-3B model with 4-bit quantization, broken down into cached Hessians and model weights (Table 19). While every methods require multiple matrix projections, CLoQ additionally performs SVD, which leads to higher memory usage in layers with large dimensions. Overall, QWHA achieves slightly lower memory footprint due to its efficient block-wise computational implementation of the fast Hadamard transform.

Table 19: Memory usage (GB) and component breakdown during initialization of each method.

| Method | Total Usage | Model Weight | Cached Hessian |
|--------|-------------|--------------|----------------|
| CLoQ | 12.77 | 3.04 | 7.10 |
| SSH | 11.93 | 3.04 | 7.10 |
| QWHA | 11.52 | 3.04 | 7.10 |

### D.7 INFERENCE OVERHEAD OF ADAPTERS

We investigate the inference throughput and memory usage of QWHA and CLoQ and present the results in Table 20 and Table 21, respectively. The evaluation uses a prefill length of 2048 and a generation length of 64, with batch size 128. We compare the FP16 pre-trained model, a quantized model with LoRA (corresponding to CLoQ), a sparse adapter (SHiRA), and FT-based adapters including WHA (QWHA), DCA (LoCA), and DHA (SSH).

As stated in SHiRA, sparse adapters provide a slight speedup over LoRA due to their simple scatter operations, compared to the low-rank matrix multiplications in LoRA. WHA introduces an additional WHT operation, but due to the fast Hadamard kernel, its overhead remains small: only a 1.9% decrease in throughput compared to LoRA. In contrast, conventional FT-based adapters such as DCA and DHA incur substantial overhead, showing a 50.9% throughput drop. Therefore, although QWHA applies an inverse WHT during inference, its overhead is marginal compared to LoRA, whereas other FT-based adapters experience significant efficiency degradation.

Regarding the inference memory usage, QWHA reduces overall memory usage by 13.0% compared to CLoQ, while both methods use 3.04 GB for model weights. This improvement arises from the use of a sparse adapter with efficient scatter operations. In addition, because the inverse WHT in QWHA is implemented without heavy matrix multiplications, it incurs no additional memory overhead and in fact results in lower peak memory usage. Consequently, QWHA achieves more than a 10% reduction in total memory consumption.

Table 20: Inference throughput (tokens/sec) of pretrained and quantized models of each adapters.

| Method | Pre-trained | LoRA | Sparse | WHA | DCA/DHA |
|---|---|---|---|---|---|
| Throughput (tokens/sec) | 66.7 | 188.1 | 191.9 | 184.6 | 92.4 |

Table 21: Peak memory usage (GB) for each method.

| Method | QWHA | CLoQ |
|---|---|---|
| Memory Usage (GB) | 52.68 | 59.53 |

### D.8 TRAINING CURVE

We analyzed the training loss, gradient norms, and convergence behavior throughout fine-tuning, and present the training curves and adapter gradient norms in Figure 10. We observe similar convergence behavior of QWHA and CLoQ. The gradient norms remain on a comparable scale despite QWHA's large nominal scaling factor. This is because the effective scaling factor of both QWHA and CLoQ is close to 1.0. Moreover, under the same effective scaling value, low-rank adapters involve matrix multiplications during backpropagation, which naturally downscales the gradient norms applied to their parameters. In contrast, the sparse adapter in QWHA does not undergo this process, resulting in gradient norms that remain stable and consistently about twice as large throughout training.

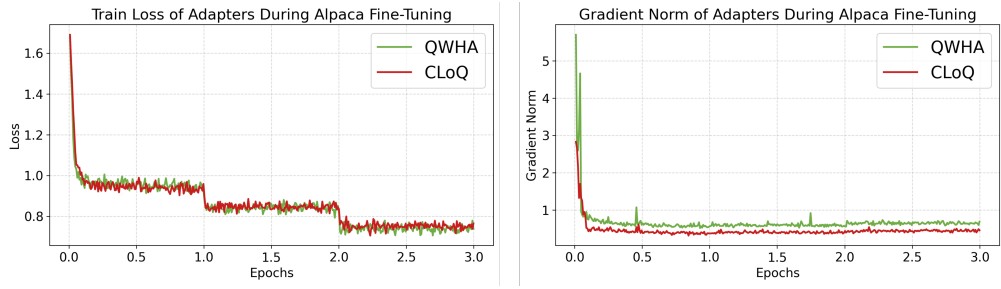

Figure 10: Training loss (left) and gradient norm (right) of each methods during Alpaca fine-tuning in LLaMA-3.2-3B 4-bit model.

