# OpenReview forum: "QWHA: Quantization-Aware Walsh-Hadamard Adaptation for Parameter-Efficient Fine-Tuning on Large Language Models"
_ICLR.cc/2026/Conference — ICLR 2026 Poster_

### Official Review · Reviewer_oADz · 2025-10-31

**Soundness:** 4
**Presentation:** 4
**Contribution:** 3
**Rating:** 8
**Confidence:** 4

**Summary:**

The paper introduces QWHA, a quantization-aware PEFT framework that integrates a Walsh–Hadamard Transform (WHT)-based adapter (WHA) with a quantization-aware initialization scheme. Unlike existing LoRA-based QA-PEFT methods that rely on low-rank updates or Fourier-transform (DCT/DHT) adapters, the proposed WHA uses a single real-valued WHT kernel with +-1 entries, reducing computation while maintaining high representational capacity. The initialization combines channel-wise parameter allocation (AdaAlloc) and value refinement to minimize layer output error caused by quantization. Experiments on LLaMA and Mistral models across 2–4 bit quantization demonstrate improved accuracy and faster training compared to LoRA and other FT-based baselines.

**Strengths:**

* Addresses a well-defined and practically relevant gap, integrating high-capacity transform adapters into quantized fine-tuning.
 * The WHT choice is analytically and empirically grounded, showing improved energy concentration and better outlier reconstruction compared to sinusoidal transforms.
 * The decomposition into adaptive allocation and refinement provides a tractable solution to an otherwise NP-hard sparse approximation problem.
 * Evaluations across multiple model families and quantization levels are comprehensive, with consistent gains in accuracy and efficiency.
 * Single-transform WHT substantially lowers compute cost, supported by observational runtime and memory measurements.

**Weaknesses:**

* Fine-tuning is limited to Alpaca (instruction-following) and general QA benchmarks; transferability to larger or domain-specific datasets remains unclear.
 * Only GPTQ+MagR quantization is used; robustness to other quantization backends is not shown.
 * While the paper reports overall training runtimes, SVD-based Hessian calibration step may still be costly for larger models and may need a breakdown for the initialization pipeline and its scaling with model size.

**Questions:**

* How sensitive is the allocation to the temperature $t$ and to very small budgets (e.g., $P \left( r \right) \leq 8$))? Any observed failure modes where some channels starve despite the per‑channel budgeting?
 * This is related to one of the weaknesses: can the proposed quantization-aware initialization generalize to other transform bases (e.g., random orthogonal, wavelet) or to rotation-based quantization schemes (e.g., AWQ, QuIP#)?

---

> ### Author Response · Authors · 2025-11-21
>
> Thank you for your constructive comments. We appreciate the insightful questions and comments you have provided.
>
> >### **Response to Weakness 1**
>
> Regarding the generalizability of QWHA in generative tasks, we note that, in addition to the instruction-following dataset (Alpaca52k), **QWHA was also fine-tuned on the mathematical reasoning dataset GSM8k and evaluated on a generative benchmark using a zero-shot CoT GSM8k prompt.**
>
> As described in Section 4 (line 372 of the original paper), GSM8k was mentioned as: “*For arithmetic reasoning, we adopt the GSM8k dataset and evaluate zero-shot accuracy following [1].*” However, it was not explicitly stated that GSM8k was also used for fine-tuning and also evaluated on a generative task, which may have led to the misunderstanding that QWHA was trained only on Alpaca52k and evaluated only on question-answering tasks. We have clarified this point in the updated paper.
>
> Our experiments therefore extend beyond instruction-following tasks and demonstrate QWHA’s effectiveness across different domains. Specifically: (a) fine-tuning on Alpaca52k dataset evaluates general question-answering ability (e.g., CSQA scores), and (b) Fine-tuning on GSM8k dataset evaluates mathematical reasoning and numerical understanding through zero-shot CoT reasoning.
>
> The results on GSM8k confirm that QWHA maintains strong reasoning performance even in domains that require multi-step problem-solving and mathematical comprehension, demonstrating that QWHA generalizes effectively beyond instruction-following datasets.
>
> [1] Training Verifiers to Solve Math Word Problems, https://arxiv.org/abs/2110.14168.

---

> ### Author Response · Authors · 2025-11-21
>
> >### **Response to Weakness 2**
>
> To address the concern that our experiments originally included only GPTQ+MagR, we conducted additional fine-tuning experiments under several distinct quantization backends, including the rotational quantization scheme SpinQuant and the simple round-to-nearest (RTN) method.
>
> Under the more challenging conditions than our main experiments, where both SpinQuant (without MagR calibration) and naive RTN introduce substantial quantization error, CLoQ often largely fails to recover accuracy in sub-4-bit settings. In contrast, QWHA with quantization-aware initialization consistently recovers the accuracy drop, demonstrating strong robustness across diverse and more difficult quantization configurations.
>
> *Caption: GSM8k fine-tuning accuracy (%) of CLoQ, QWHA and QWHA without initialization on LLaMA-3.2-3B under different quantization backends.*
>
> | Method | SpinQ | RTN |
> |---|---|---|
> | CLoQ | 40.41 | 40.51 |
> | QWHA | 41.92 | 41.24 |
> | CLoQ | 36.39 | 19.33 |
> | QWHA | 36.92 | 37.52 |
> | CLoQ | 16.02 | 0.37 |
> | QWHA | 23.88 | 16.98 |
>
> Overall, these results show that QWHA remains effective across diverse quantization schemes and provides even larger benefits in challenging settings with substantial quantization error.

---

> ### Author Response · Authors · 2025-11-21
>
> >### **Response to Weakness 3**
>
> For the initialization cost of the adapters, we present a detailed breakdown of the initialization time for QWHA and CLoQ in the table below. A summarized version also appears in the updated paper (Appendix D.5, Table 15).
>
> *Caption: Initialization latency (hours) for each model and breakdown of computation steps.*
>
> | Model | Method | Hessian Computation | Adapter Initialization | Total |
> |---|---|---|---|---|
> | LLaMA-3.2-3B | CLoQ | 0.44 | 0.14 | 0.58 |
> |  | SSH | 0.44 | 0.45 | 0.89 |
> |  | QWHA | 0.44 | 0.22 | 0.66 |
> | LLaMA-3.1-8B | CLoQ | 0.90 | 0.23 | 1.14 |
> |  | SSH | 0.90 | 0.96 | 1.86 |
> |  | QWHA | 0.90 | 0.43 | 1.34 |
> | Mistral-7B-v0.3 | CLoQ | 1.02 | 0.24 | 1.26 |
> |  | SSH | 1.02 | 0.95 | 1.98 |
> |  | QWHA | 1.02 | 0.43 | 1.46 |
>
> As described in Section 3, the initialization process of QWHA consists of two steps: (a) gathering activations and quantization errors, and (b1) selecting parameters and refining their values using WHT-based transforms and solving a channel-wise least-squares problem. Similarly, CLoQ also consists of two steps:(a) gathering activations and quantization errors, and (b2) applying SVD before solving the least-squares problem (as described on page 6, Equation (5) of CLoQ [1]).
>
> Thus, the main practical difference in initialization latency between CLoQ and QWHA arises in step (b1) versus (b2). This difference is marginal, only around ten minutes for 7-8B models, while the majority of the initialization time for both methods is dominated by Hessian computation. Although FT-based adapters include additional matrix projections, the use of the fast WHT kernel in step (b1) allows QWHA to maintain overall initialization time comparable to CLoQ. By contrast, other FT-based adapters incur significantly higher latency because their transform kernels are slower. Moreover, initialization time is small relative to the full fine-tuning duration, so the end-to-end runtime of QWHA remains similar to CLoQ.
>
> In summary, Hessian computation is the primary cost driver for activation-aware initialization in both QWHA and CLoQ. Beyond this shared cost, QWHA shows initialization latency comparable to CLoQ and substantially lower latency than other FT-based adapters. Consequently, the complete fine-tuning pipeline of QWHA retains efficiency comparable to CLoQ, with far lower overhead than conventional FT-based methods.
>
> [1] *CLoQ: Enhancing Fine-Tuning of Quantized LLMs via Calibrated LoRA Initialization*, https://arxiv.org/pdf/2501.18475

---

> ### Author Response · Authors · 2025-11-21
>
> >### **Response to Question 1**
>
>
> To evaluate the sensitivity of QWHA’s parameter allocation to the temperature parameter   $   t   $  , we conducted additional experiments analyzing the rank of the adapter after initialization, which reflects whether parameters are sufficiently distributed across channels without depriving any channel of the capacity needed for effective fine-tuning. As shown in the table below, the normalized rank of the initialized QWHA weights remains consistently high across all projection types, with variations below 0.2%. This demonstrates that QWHA’s allocation strategy remains robust as long as   $   t   $   lies within a reasonable range.
>
> *Caption: Normalized rank (%) of linear projections initialized with different temperatures on the 4-bit quantized LLaMA-3.2-3B model with   $   P(r=64)   $  .*
>
> | Temperature (t) | 0.25 | 0.5 | 1 | 1.5 | 2 |
> |-----------------|------|------|------|------|------|
> | Query           | 99.87 | 99.87 | 99.86 | 99.83 | 99.79 |
> | Key             | 100.00 | 100.00 | 100.00 | 100.00 | 100.00 |
> | Value           | 100.00 | 100.00 | 100.00 | 100.00 | 100.00 |
> | Out             | 99.72 | 99.71 | 99.66 | 99.60 | 99.57 |
> | Up              | 100.00 | 100.00 | 100.00 | 100.00 | 100.00 |
> | Gate            | 100.00 | 100.00 | 100.00 | 100.00 | 100.00 |
> | Down            | 100.00 | 100.00 | 100.00 | 100.00 | 100.00 |
>
> To further address the reviewer’s concern regarding potential *channel starvation* under extremely small parameter budgets, we evaluated QWHA at lower   $   r   $   values. As shown below, when   $   r = 8   $  , two projection types (query and out) exhibit reduced rank due to their highly peaked per-channel weight distributions under such limited parameter budgets. However, this behavior is confined to these extremely small settings and disappears quickly as   $   r   $   increases, indicating that the issue arises from aggressive parameter reduction rather than from QWHA itself.
>
> *Caption: Average normalized rank (%) of QWHA applied to LLaMA-3.2-3B over parameter budgets.*
>
> | Projection type | P(r=8) | P(r=16) | P(r=32) | P(r=64) |
> |-----------------|---------|---------|---------|---------|
> | Query           | 97.22 | 99.38 | 99.80 | 99.86 |
> | Key             | 100.00 | 100.00 | 100.00 | 100.00 |
> | Value           | 100.00 | 100.00 | 100.00 | 100.00 |
> | Out             | 73.44 | 88.79 | 97.21 | 99.66 |
> | Up              | 100.00 | 100.00 | 100.00 | 100.00 |
> | Gate            | 100.00 | 100.00 | 100.00 | 100.00 |
> | Down            | 100.00 | 100.00 | 100.00 | 100.00 |
>
> Overall, these results show that QWHA is highly robust to the temperature parameter and reliably maintains full-rankness across practical parameter budgets.

---

> ### Author Response · Authors · 2025-11-21
>
> >### **Response to Question 2**
>
> The proposed quantization-aware initialization is not limited to FT-related transforms. It can be applied to any type of sparse adapter constructed over an invertible transform basis, as long as the inverse transform is available. This includes random orthogonal transforms, wavelet-based transforms, and other related bases. Components of our initialization method (AdaAlloc and the Refinement step) can be used directly in these settings. Among the transform families we explored, WHT proved the most effective due to its strong energy-concentrative coefficient distribution.
>
> In addition, our initialization method is compatible with a wide range of quantization schemes, including rotation-based methods. Across different quantization backends and their associated error distributions, WHT remains effective at capturing and correcting large-magnitude quantization errors, which are the primary contributors to accuracy degradation. As discussed in the Response to Weakness 2, QWHA shows clear improvements even under SpinQuant, a rotational quantization scheme, demonstrating that our initialization approach generalizes well to rotation-based quantization settings.

---

### Official Review · Reviewer_L6oQ · 2025-10-31

**Soundness:** 2
**Presentation:** 3
**Contribution:** 3
**Rating:** 4
**Confidence:** 3

**Summary:**

This paper introduces QWHA, a novel Quantization-Aware Parameter-Efficient Fine-Tuning (QA-PEFT) method. QWHA utilizes a Walsh-Hadamard Transform (WHT)-based adapter, termed WHA, designed to effectively mitigate quantization errors in Large Language Models (LLMs). The core contributions are a computationally efficient adapter design using WHT, and a sophisticated initialization scheme that combines adaptive parameter allocation (AdaAlloc) to preserve rank and target high-error channels, with a value refinement step to minimize initial error. Experimental results on LLaMA and Mistral models demonstrate that QWHA outperforms existing QA-PEFT methods in low-bit settings (2, 3, and 4-bit) and is significantly faster to train than other Fourier-related transform (FT)-based adapters.

**Strengths:**

- **Novel and Well-Motivated Adapter Design (WHA)**: A key strength of this work is the innovative use of the Walsh-Hadamard Transform (WHT) for the adapter, and what makes this choice particularly strong is its tight alignment with the inherent nature of quantization errors. From Section 3.1, we can see that the authors compellingly argue WHT’s square-wave basis functions are far better suited to capturing the abrupt, outlier-driven characteristics of quantization errors than the smooth sinusoidal bases of DCT/DHT employed in prior Fourier-based adapters—and this claim is not just theoretical, as demonstrated in Figure 2(b)’s energy concentration analysis, which clearly validates the superiority of WHT in this context. Additionally, the decision to adopt a single transform (instead of the dual transforms used in earlier work) is both theoretically sound and practically impactful: Section 3.1’s discussion of quantization error’s channel independence justifies the simplification, while Table 13 further shows that this design delivers significant training speedups without compromising performance, making the method much more viable for real-world applications.

- **Effective Quantization-Aware Initialization Strategy**: The proposed two-stage initialization scheme stands out as a notable strength, with its effectiveness evident across Section 3.2, Algorithm 1, Figure 4, Table 2, and Figure 5. AdaAlloc (Eq. 7) offers a principled heuristic that skillfully balances two critical objectives: reducing quantization error by prioritizing high-error channels and preserving the model’s fine-tuning capacity by maintaining a high-rank adapter structure—as illustrated in Figure 4. This is a clear advancement over simplistic magnitude-based or random parameter selection strategies. Complementing this, the subsequent Refinement step (Eq. 9) significantly reduces initial layer output error, as we can see from Table 2 and Figure 5, providing a far stronger starting point for fine-tuning than conventional initialization approaches.

- **Strong and Consistent Empirical Performance**: The method’s performance advantages over existing baselines prove to be robust and reliable, as Table 3, Table 4, and Figure 6 consistently show. Across multiple models (LLaMA-3.1-8B, LLaMA-3.2-3B, Mistral-7B-v0.3) and different bit-widths, QWHA demonstrates clear and consistent superiority. Most impressively, in the highly challenging 2-bit quantization setting, as seen in Table 3, the method achieves a 2-4% absolute improvement on CSQA and GSM8k tasks compared to the next-best baseline—highlighting its exceptional effectiveness in extreme compression scenarios. Figure 6 further reveals that QWHA outperforms CLoQ even at its peak performance, despite using a smaller parameter budget (r=32), which underscores that the proposed adapter’s strength stems from superior representational capacity rather than mere efficient parameter allocation.

**Weaknesses:**

- **Insufficient Hyperparameter Sensitivity Analysis**: From Section 4, Table 8, Table 9, and Equation (7), it’s clear the study relies heavily on several critical hyperparameters without exploring their sensitivity—a notable oversight for a method emphasizing practicality. Notably, the AdaAlloc temperature `t` is fixed at 1 with no ablation to show how adjusting it might alter the parameter allocation strategy (e.g., making it more or less focused on high-error channels). Even more concerning, the adapter scaling factor `α` is set to 4000 for QWHA versus just 1 for CLoQ (per Table 8)—a massive discrepancy that goes entirely unjustified. This raises questions about whether the performance gains stem from the method itself or simply favorable hyperparameter tuning. Additionally, the learning rates in Table 9 appear to be heavily tailored to each experimental setting, which casts doubt on how easily QWHA can be applied to new models or tasks without extensive retuning.

- **Limited Scope of Experimental Evaluation**: The experimental evaluation, as outlined in Section 4’s “Models and Datasets” section, is overly narrow, which undermines the claim of QWHA’s generalizability. Currently, all results are based on instruction tuning with the Alpaca dataset, followed by evaluation on just two tasks: commonsense QA (CSQA) and arithmetic reasoning (GSM8k). While these are relevant benchmarks, they do not capture the full range of use cases for quantized LLMs. It would greatly strengthen the paper to include results on more diverse tasks—such as text summarization (e.g., XSum), code generation (e.g., HumanEval), or long-context understanding—to demonstrate that QWHA’s benefits extend beyond reasoning-focused tasks. Without this, it’s hard to assess whether the method is broadly applicable or limited to specific task types.

- **Lack of Statistical Rigor in Results**: Looking at Tables 3, 4, 10, and 11, it appears all reported accuracy scores are from single runs—an issue that contradicts standard practices in top-tier ML conferences. Fine-tuning LLMs is known to exhibit significant variability based on random seeds, and without reporting mean values and standard deviations across multiple runs, it’s impossible to determine if the reported performance gains are statistically significant or merely due to chance. This lack of rigor makes it difficult to trust the consistency of QWHA’s results, especially in the closely contested low-bit settings where small differences can tip the balance between methods.

- **Inconsistent or Unfair Baseline Comparisons**: From Table 3 and Section 2.3, it’s apparent that the primary performance comparison is somewhat inconsistent. QWHA, which benefits from a dedicated quantization-aware initialization scheme, is pitted against other Fourier-based adapters (LoCA, SSH) that lack this critical component. While the paper acknowledges this mismatch, it still frames this as the main comparison—creating a potentially unfair advantage for QWHA. The more apples-to-apples comparison in Table 4 (where AdaAlloc is applied to all adapter types) paints a more nuanced picture: DCA + AdaAlloc and DHA + AdaAlloc often perform competitively with WHA, especially in the 2-bit LLaMA-3.2-3B GSM8k setting. This complicates the narrative that WHA’s transform kernel is the key differentiator, suggesting the initialization scheme may play a more dominant role than claimed.

- **Weaknesses in Mathematical Formulation and Notation**: Several clarity issues in the mathematical presentation, visible in Algorithm 1, Appendix B.2, and Appendix C.1, hinder reproducibility and understanding. In Algorithm 1, the notation `(p_0, ..., p_d_out) ∈ N_d_out` is non-standard; using `(p_0, ..., p_{d_{out}-1}) ∈ ℕ^{d_{out}}` would make the dimensioning far clearer for readers. Additionally, the justification for WHA’s full-rank property in Appendix B.2 relies on theory for *random* sparse matrices [Coja-Oghlan et al., 2020], but AdaAlloc is a deterministic parameter allocation scheme—creating a tenuous link between the theoretical support and the actual method. While the derivation in Appendix C.1 is mathematically correct, it could be more self-contained by explicitly defining `U` and `Σ` as components of the SVD of `XXᵀ`, rather than assuming readers will infer this.

- **Limited Ablation on Core Method Components**: The paper’s ablation studies, while present, fail to address key questions about its core design choices—something evident from Figure 5, Table 7, and Table 13. For example, the Refinement step’s impact is only measured on *initialization error* (per Figure 5 and Table 7), not on final *task accuracy*. An ablation showing how “WHA + AdaAlloc” performs without Refinement would be critical to understanding whether this step adds meaningful value beyond reducing early-stage error. Similarly, Table 13 compares training times for 1D versus 2D transforms but provides no corresponding accuracy data. Since the use of a single transform is a central design choice, empirical proof that a 2D transform offers no accuracy benefit (not just speed drawbacks) is necessary to justify this decision.

- **Generalization to Other Quantization Schemes**: Despite claiming QWHA is compatible with any quantization scheme, the paper only presents results using GPTQ [Frantar et al., 2023], as seen in Section 4’s “Baselines” section. This gap between claim and evidence weakens the argument for QWHA’s versatility. Quantization methods vary significantly in how they handle outliers, weight distribution, and compression logic—for example, AWQ [Lin et al., 2024] uses activation-aware weighting, while RTN relies on simple rounding. Demonstrating QWHA’s effectiveness with at least one other popular scheme would go a long way toward proving its generalizability, rather than limiting it to a single quantization pipeline.

- **Clarity on Parameter Budget (`r`)**: From Sections 3 and 4, it’s apparent that the paper uses the LoRA-specific term “rank (`r`)” to define the parameter budget for *all* adapters (e.g., “P(r=64)”)—a choice that creates unnecessary confusion. For Fourier-based adapters like WHA, `r` does not have a direct “rank” interpretation, as their structure differs fundamentally from LoRA’s low-rank matrices. Using a more generic term (e.g., “budget size” or “trainable parameter count”) and explicitly reporting the total number of trainable parameters for each method would greatly improve clarity, especially for readers comparing QWHA to non-LoRA baselines.

- **Inference Overhead Analysis is Missing**: Unfortunately, there is no discussion of inference overhead anywhere in the manuscript—an odd omission for a paper that emphasizes efficiency. While the authors likely intend to merge the adapter weights `ΔW` into the frozen quantized weights `W_Q` for zero runtime overhead, the one-time cost of computing `ΔW = F H⁻¹` (including the inverse WHT and sparse-dense matrix multiplication) is never addressed. Additionally, the memory footprint of the sparse `F` matrix compared to LoRA’s compact `A` and `B` matrices is left unmentioned. To complete the efficiency narrative, a brief analysis of these inference-related costs would be essential for practitioners deciding whether to adopt QWHA.

**Questions:**

1. Regarding the adapter scaling factor `α` (Table 8), there is a massive discrepancy between QWHA (α=4000) and CLoQ (α=1)—could you elaborate on why this choice was made? Was `α` tuned independently for each method, and if so, have you investigated how such a large value impacts gradient norms, training stability, or convergence speed for QWHA?

2. The main comparison in Table 3 pairs QWHA (with its dedicated quantization-aware initialization) against other Fourier-based methods that lack this feature, while Table 4 provides a more controlled comparison by applying AdaAlloc to all adapter types. In Table 4, `DCA + AdaAlloc` and `DHA + AdaAlloc` perform quite competitively with WHA—particularly in the 2-bit LLaMA-3.2-3B GSM8k setting. Could you comment on this? Does this suggest that the initialization scheme (AdaAlloc) is the dominant driver of performance, rather than the specific choice of transform kernel (WHT)?

3. The justification for using a single transform (instead of dual transforms) is compelling from an efficiency standpoint, and Table 13 clearly shows the training time benefits. However, to fully validate this design choice, could you provide an accuracy comparison between 1D WHA and a hypothetical 2D WHA? This would empirically confirm that the second transform offers no meaningful accuracy gain in the QA-PEFT context, rather than just faster training.

4. The AdaAlloc method uses a fixed temperature parameter `t=1` (Section 4), but there is no sensitivity analysis provided. Have you tested how varying `t` (e.g., t=0.5, t=2) affects parameter allocation sharpness and final performance? For instance, does a lower `t` lead to over-concentration on a small number of channels, or a higher `t` reduce effectiveness by spreading parameters too evenly?

5. As noted earlier, the reported results lack error bars or confidence intervals from multiple runs. Given the known variance in LLM fine-tuning due to random seeds, could you comment on the stability of QWHA’s performance across different seeds? Have you observed consistent gains, or is there significant variability that might affect the reliability of the reported results?

---

> ### Author Response · Authors · 2025-11-20
>
> Thank you for your constructive comments. We appreciate the insightful questions and comments you have provided.
>
> >### **Response to Weakness 1 (part 1)**
>
> - **Temperature parameter $ t $:**
>
> Regarding the choice of the temperature parameter $ t $, we aimed to select a value that preserves the full-rankness of the adapter weights for effective fine-tuning, while still being high enough to emphasize high-error channels and maximize the effectiveness of quantization-aware initialization. We conducted additional experiments evaluating the sensitivity of QWHA to $ t $ in terms of both parameter allocation (full rankness) and fine-tuned accuracy, and found that $ t = 1 $ satisfies these criteria well.
>
> For parameter allocation, we measure the rank of the adapter after initialization, which reflects whether parameters are sufficiently distributed across channels without depriving any channel of the capacity needed for effective fine-tuning. As shown in the table below, the rank of the initialized QWHA weights remains consistently high across all projection types, with variations below 0.2%. This indicates that the resulting parameter allocation for effective fine-tuning remains robust as long as $ t $ is within a reasonable range.
>
> *Caption: Normalized rank (%) of linear projections initialized with different temperatures on the 4-bit quantized LLaMA-3.2-3B model with $ P(r=64) $.*
>
> | Temperature (t) | 0.25 | 0.5 | 1 | 1.5 | 2 |
> |-----------------|-------|------|------|-------|------|
> | Query           | 99.87 | 99.87 | 99.86 | 99.83 | 99.79 |
> | Key             | 100.00 | 100.00 | 100.00 | 100.00 | 100.00 |
> | Value           | 100.00 | 100.00 | 100.00 | 100.00 | 100.00 |
> | Out             | 99.72 | 99.71 | 99.66 | 99.60 | 99.57 |
> | Up              | 100.00 | 100.00 | 100.00 | 100.00 | 100.00 |
> | Gate            | 100.00 | 100.00 | 100.00 | 100.00 | 100.00 |
> | Down            | 100.00 | 100.00 | 100.00 | 100.00 | 100.00 |
>
> For fine-tuned model accuracy, the GSM8k results shown below (also included as Table 13 in Appendix D.3) exhibit very small variation: within 0.08%p for $ t \in [0.5, 1.0] $, and within 1.5%p across the entire tested range. We note that while low $ t $ spreads parameters too uniformly across channels, under-allocating to channels with large quantization error, high $ t $ over-concentrates parameters in high-error channels while neglecting important coefficients inside low-magnitude channels. As a result, both excessively high and excessively low temperatures lead to drops in accuracy.
>
> *Caption: GSM8k Accuracy (%) of QWHA models initialized with different temperature values on the 4-bit quantized LLaMA-3.2-3B model with $ P(r=64) $.*
>
> | Temperature | 0.25 | 0.5 | 1 | 1.5 | 2 |
> |-------------|------|------|------|------|------|
> | GSM8k Acc   | 40.11 | 41.39 | 41.47 | 40.64 | 40.04 |
>
> As a result, within the robust range of $ t \in [0.5, 1.0] $, we adopt $ t = 1 $ as the default setting, as it naturally prioritizes outlier-affected, high-magnitude channels while maintaining stable empirical performance.
>
> We acknowledge that the original submission omitted this ablation and have added the results and further explanations to Appendix D.3 of the revised paper for better clarity and visibility.

---

> ### Author Response · Authors · 2025-11-20
>
> >### **Response to Weakness 1 (part 2)**
>
> - **Adapter scaling factor  $ \alpha $**
>
> Regarding the scaling factor of QWHA, we emphasize that the seemingly large value of  $ \alpha = 4000 $  was not chosen to enhance fine-tuning performance. Instead, it compensates for an implementation-specific rescaling in our code and ensures that QWHA uses the same *effective* scaling as CLoQ (namely,  $ \alpha_{ effective } \approx 1.0 $ ).
>
> In our current implementation, which can be found in the submitted code’s QWHA layer definition, the explicitly specified scaling factor is internally divided by the layer input dimension  $ d_{ in } $ , similar to how the original LoRA implementation divides its scaling factor by the rank as  $ \alpha_{ effective} = \alpha_{ explicit} / r $ . We updated the low-rank adapter code so that it directly uses an effective scaling factor of 1.0 as  $ \alpha_{effective} =1 $ , but we unintentionally did not apply the same fix to QWHA. As a result, the actual scaling used by QWHA becomes  $ \alpha_{ effective } = \alpha_{explicit} / d_{in} $ .
>
> To correct this behavior, we set  $ \alpha_{ explicit } = 4000 $ , which is close to the typical input dimensions of our models (3072 for LLaMA-3.2-3B and 4096 for LLaMA-3.1-8B and Mistral-7B-v0.3). Under this implementation, the effective scaling factor is   $ \alpha_{ effective } \approx 4000 / d_{ in } \approx 1.0 $ , which matches the  $ \alpha = 1.0 $  setting used in CLoQ and ensures consistent gradient scaling across methods.
>
> Thus, the numerical choice of  $ \alpha_{ explicit } = 4000 $  serves only to restore the intended effective scaling. Therefore, QWHA’s accuracy gains are the improvements that arise from the adapter design and the quantization-aware initialization rather than the seemingly large scaling factor.
>
> We briefly noted in Appendix D.1 as: “The scaling factors for low-rank and FT-based adapters are defined so that the  $  \ell_2  $  norm of the gradients applied to their respective parameters is aligned.”, but we agree that the description was largely unclear. In the revised paper, we explicitly clarify the distinction between  $ \alpha_{ explicit } $  and  $ \alpha_{ effective } $  and plan to update the implementation so that the provided scaling factor is used directly, without unintended division by  $ d_{ in } $ . We appreciate the reviewer’s careful attention to this.
>
> - **Training behavior of QWHA:**
>
> We further analyzed the training loss, gradient-norm trajectories, and convergence behavior throughout fine-tuning, and report these results in Figure 10 of the updated Appendix D.8. The training losses follow similar patterns across methods, and QWHA’s gradient norms remain on a comparable scale to those of CLoQ. This is because both methods ultimately operate under an *effective* scaling factor close to  $ 1.0 $ . Additionally, low-rank adapters inherently undergo matrix multiplications during backpropagation, which naturally downscale their gradients, whereas the sparse adapter in QWHA does not undergo this process. As a result, QWHA’s gradient norms remain roughly twice as large throughout training, yet still well-converged.
>
> - **Learning rate of each method**
>
> For CLoQ, we applied a similar grid-search procedure used for QWHA and selected the best-performing learning rate, while keeping all other hyperparameters aligned with those in the original CLoQ paper. We found that learning rates in the range of   $  2 \times 10^{-5}  $  to  $  2 \times 10^{-4}  $  , together with an effective scaling factor close to  $ 1.0 $ , yield a stable and well-converged models for both CLoQ and QWHA.
>
> ---
>
> Overall, these observations confirm that QWHA’s performance and convergence behavior do not arise from extensive hyperparameter tuning, but rather from the principled design of the adapter and its quantization-aware initialization.

---

> ### Author Response · Authors · 2025-11-20
>
> >### **Response to Weakness 2**
>
> Regarding the generalizability of QWHA in generative tasks, we note that, in addition to the instruction-following dataset (Alpaca52k), **QWHA was also fine-tuned on the mathematical reasoning dataset GSM8k and evaluated on a generative benchmark using a zero-shot CoT GSM8k task.**
>
> As described in Section 4 (line 372 of the original paper), GSM8k was mentioned as: “*For arithmetic reasoning, we adopt the GSM8k dataset and evaluate zero-shot accuracy following [ref].*” However, it was not explicitly stated that GSM8k was also used for fine-tuning as well as evaluation, which may have led to the misunderstanding that QWHA was trained only on Alpaca52k. We have clarified this point in the updated paper.
>
> Our experiments therefore extend beyond instruction-following tasks and demonstrate QWHA’s effectiveness across different domains. Specifically: (a) fine-tuning on Alpaca52k dataset evaluates general question-answering ability (e.g., CSQA scores), and (b) Fine-tuning on GSM8k dataset evaluates domain-specific mathematical reasoning and numerical understanding through zero-shot CoT reasoning. The results on GSM8k confirm that QWHA maintains strong reasoning performance even in domains that require multi-step problem-solving and mathematical comprehension, demonstrating that QWHA generalizes effectively beyond instruction-following datasets.
>
> In addition, these two configurations have been the most commonly used evaluation pipeline in recent QA-PEFT works. However, we agree that incorporating a broader range of datasets would further strengthen our evaluation. Therefore, we plan to extend QWHA to a summarization dataset such as XSum or a code generation dataset such as HumanEval in future experiments as resources allow.

---

> ### Author Response · Authors · 2025-11-20
>
> >### **Response to Weakness 3**
>
> To further assess the robustness of QWHA, we conducted additional experiments evaluating stability across multiple random seeds. Using five new seeds (1111 to 5555), in addition to the originally reported seed of 42, we report the accuracy differences relative to the original result on the LLaMA-3.2-3B Alpaca fine-tuning setup, along with the standard deviation for each bit-width, in the table below.
>
> *Caption: Differences and Standard Deviations (denoted as Std.) of QWHA Accuracy (%) Across Random Seeds*
>
> | Bit-width | 1111   | 2222   | 3333   | 4444   | 5555   | Std.|
> |-----------|--------|--------|--------|--------|--------|-----------|
> | 2-bit     | -0.130 | 0.046  | -0.040 | -0.084 | -0.050 | 0.0648    |
> | 3-bit     | 0.177  | -0.054 | 0.057  | 0.290  | 0.006  | 0.1383    |
> | 4-bit     | -0.081 | 0.017  | -0.050 | 0.077  | -0.011 | 0.0613    |
>
> Across all bit-widths, the variations remain small, and importantly, QWHA consistently stays above all baseline methods under these fluctuations. These results indicate that the performance improvements provided by QWHA are stable and robust across different random seeds.

---

> ### Author Response · Authors · 2025-11-20
>
> >### **Response to Weakness 4**
>
> - **Baseline comparisons in Table 3**
>
> We understand the reviewer’s concern that Table 3 compares QWHA, which includes a dedicated quantization-aware initialization method, against prior FT-based adapters (LoCA, SSH) that were originally designed for high-precision models and therefore do not incorporate such initialization. However, this comparison is not intended as an omission, but rather reflects a core contribution of our work. To the best of our knowledge, no prior method provides a quantization-aware initialization procedure for FT-based adapters, largely because selecting and optimizing sparse spectral coefficients for quantized weights is a non-trivial problem. QWHA is the first to introduce such an initialization strategy by heuristically solving the Sparse Approximation Problem. Since QWHA’s contribution comprises both (1) the WHA transform kernel and (2) the quantization-aware initialization (AdaAlloc + Refinement), Table 3 evaluates the complete QWHA pipeline to demonstrate the effectiveness of the full method.
>
> To provide a more controlled comparison, Table 4 applies our initialization scheme uniformly to all FT-based adapters (DCA, DHA, WHA), and also compares multiple initialization variants within WHA. This breakdown shows that both the transform kernel and the initialization play meaningful roles.
>
> - **Adapter and initialization comparisons in Table 4**
>
> As noted in the reviewer’s comment, the initialization process significantly boosts model accuracy, which naturally narrows the differences among adapters when all use the same initialization. Nonetheless, Table 4 indicates that WHA fully benefits from AdaAlloc. In all settings, DCA or DHA paired with AdaAlloc underperform relative to WHA, and in some cases even fall below the low-rank CLoQ baseline (e.g., 3-bit GSM8k on LLaMA-3.2-3B). This suggests that while initialization is indeed an important component, the transform kernel is still critical: WHA’s energy-concentrative coefficient distribution enables AdaAlloc to operate at full effectiveness, whereas other transforms struggle to approximate quantization errors even under identical initialization.
>
> Additionally, WHA offers advantages beyond accuracy. Its WHT-based kernel enables significantly faster computation than DCT- or DHT-based adapters, as shown in Section 4.2. This efficiency contribution is unique to WHA and therefore we highlighted it as a *key factor to the training efficiency.*

---

> ### Author Response · Authors · 2025-11-20
>
> >### **Response to Weakness 5**
>
> - **Typo in the Algorithm 1:**
>
> Thank you for pointing out the inconsistency. We have corrected the channel-wise notation in Algorithm 1 from  $ (p_0, \ldots, p_{d_{out}}) $  to the standard and dimensionally correct form  $ (p_0, \ldots, p_{d_{out}-1}) \in \mathbb{N}^{d_{out}} $ .
>
> We appreciate the reviewer’s careful attention to detail.
>
> - **Full-rank property of AdaAlloc:**
>
> Regarding the concern about AdaAlloc and the full-rank justification in Appendix B.2, we clarify that although AdaAlloc is a deterministic allocation scheme, it still satisfies the essential sufficient conditions required for a sparse matrix to be full rank.
>
> As described in Appendix B.2, the key requirement for a sparse matrix to achieve full rank is that each column and each row receives at least two nonzero entries on average. AdaAlloc’s per-channel allocation, combined with remainder assignment and temperature control, guarantees that every output channel receives at least two parameters, thereby fulfilling the column-wise condition directly. Moreover, while AdaAlloc selects coefficient indices within each channel based on the magnitude of  $  \Delta W_Q H  $ , the distribution of these WHT-transformed coefficients is nearly random normal except for a small portion of outliers (Appendix B.3, Figure 8), due to the near-zero correlation across input rows. As a result, the selected index positions along rows behave effectively like random choices.
>
> This pattern is clearly observable in Appendix C.2 (Figure 9): the vertical structure reflects AdaAlloc’s deterministic per-channel budget assignment, while the row-wise distribution of selected indices appears almost random, indicating that the row-wise coverage condition holds in practice. Therefore, even though AdaAlloc is deterministic, it satisfies the same sufficient conditions as random sparse matrices and thus ensures the full-rank property with high probability.
>
> Previously, we briefly noted this in the “practical consideration” paragraph of Appendix B.2 as “Importantly, since AdaAlloc selects indices independently within each channel, the parameters across input rows can be treated as uniformly random selections. Moreover, AdaAlloc’s per-channel allocation with remainder assignment and temperature control guarantees at least two elements in every channel.”, but we acknowledge that the explanation was overly concise. In the revised paper, we have expanded this section to more clearly address why AdaAlloc satisfies the theoretical full-rank conditions.
>
> - **Notations and definitions in Appendix C.1:**
>
> Thank you for pointing out the missing descriptions in the appendix. We have added explicit definitions of  $  U  $  and  $  \mathbf{\Sigma}  $  as the eigenvector and eigenvalue matrices, respectively, obtained from the SVD of the Hessian Gram matrix  $  X X^\top  $ .
>
> We appreciate the reviewer’s careful attention to detail.

---

> ### Author Response · Authors · 2025-11-20
>
> >### **Response to Weakness 6**
>
> To address the concern on the core design choices, we conducted additional experiments and report the final task accuracy for both (a) WHA with and without the Refinement step and (b) 1D versus 2D WHA. The results are summarized below.
>
> *Caption: Benchmark accuracy (%) according to the refinement and transform dimensionality in QWHA.*
>
> | Method | QA init. | Coeff Selection | Refine. | Adapter | 4bit |  | 3bit |  | 2bit |  |
> |---|---|---|---|---|---|---|---|---|---|---|
> |     |     |     |     |     | CSQA | GSM8k | CSQA | GSM8k | CSQA | GSM8k |
> | QWHA | O | AdaAlloc | O | 1D WHA | 66.11 | 41.47 | 64.8 | 39.58 | 57.03 | 29.11 |
> | QWHA | O | AdaAlloc | O | 2D WHA | 65.92 | 41.43 | 64.56 | 38.66 | 57.21 | 30.85 |
> | QWHA | O | AdaAlloc | X | 1D WHA | 65.58 | 39.57 | 64.4 | 37.3 | 56.08 | 28.65 |
>
> The results show that removing the Refinement step consistently reduces accuracy across all datasets and bit-widths, indicating that Refinement is not merely helpful for reducing initialization error but also yields clear improvements in final task performance.
>
> For transform dimensionality, the accuracy difference between 1D and 2D WHA is small in nearly all settings, with only a modest deviation observed in the 2-bit GSM8k case. Because 2D WHA still applies AdaAlloc in the same way and can be viewed as a natural variant of QWHA, these findings suggest that adding a second transform does not meaningfully improve accuracy in the QA-PEFT setting, while incurring additional training cost.
>
> Overall, these experiments support our design choice: Refinement plays an important and beneficial role, whereas a second transform provides little accuracy gain relative to the efficient 1D WHA.

---

> ### Author Response · Authors · 2025-11-20
>
> >### **Response to Weakness 7**
>
> We conducted additional experiments by fine-tuning QWHA under multiple quantization schemes, based on the fact that adapter initialization interacts closely with how each quantization method handles outliers. Since AWQ and GPTQ are both activation-aware quantization schemes, we additionally include RTN, which relies on simple rounding and therefore produces heavier outliers, and SpinQuant, a rotational quantization method that is activation-aware and applies weight rotations to reduce outlier effects.
>
> As shown in the table below, both SpinQuant (without MagR calibration) and naive RTN introduce substantial quantization error. Under these more challenging conditions than our main experiments, CLoQ often fails to recover accuracy in sub-4-bit settings. In contrast, QWHA with quantization-aware initialization consistently recovers the accuracy drop, demonstrating strong robustness across diverse and more difficult quantization configurations.
>
> *Caption: GSM8k fine-tuning accuracy (%) of CLoQ, QWHA and QWHA without initialization on LLaMA-3.2-3B model.*
>
> | Method | SpinQ | RTN |
> |---|---|---|
> | CLoQ| 40.41 | 40.51 |
> | QWHA| 41.92 | 41.24 |
> | CLoQ| 36.39 | 19.33 |
> | QWHA| 36.92 | 37.52 |
> | CLoQ | 16.02 | 0.37 |
> | QWHA| 23.88 | 16.98 |
>
> As a result, these results demonstrate that QWHA maintains strong performance even under challenging quantization regimes, reinforcing its generalizability beyond a single quantization pipeline.

---

> ### Author Response · Authors · 2025-11-20
>
> >### **Response to Weakness 8**
>
> Regarding the parameter budget  $ P(r) $ , our intention was to adopt a concise and interpretable notation that aligns with prior PEFT literature, rather than listing the raw parameter counts for sparse adapters (as defined in Section 3). This makes comparisons with low-rank adapters straightforward. Although the total number of trainable parameters is shown in Figures 3 and 6, we agree that explicitly clarifying the relationship between the total parameter count and  $ P(r) $  would improve clarity, especially for readers comparing QWHA with non–LoRA-style FT-based adapters.
>
> We now include the full parameter counts in the table below, which also appears as Table 10 in Appendix D.1 of the updated paper. We note that the total number of trainable parameters scales proportionally with  $ r $ , and is identical to that of a low-rank adapter with rank  $ r $ .
>
> *Caption: Total number of parameters at  $ P(r = 64) $  for each projection, layer, and model.*
>
> | Units | q_proj | k_proj | v_proj | o_proj | gate_proj | up_proj | down_proj | Per-layer | Per-model |
> |---|---|---|---|---|---|---|---|---|---|
> | LLaMA-3.1-8B | 524288 | 327680 | 327680 | 524288 | 1179648 | 1179648 | 1179648 | 5242880 | 167772160 |
> | LLaMA-3.2-3B | 393216 | 262144 | 262144 | 393216 | 720896 | 720896 | 720896 | 3473408 | 97255424 |
> | Mistral-7B-v0.3 | 524288 | 327680 | 327680 | 524288 | 1179648 | 1179648 | 1179648 | 5242880 | 167772160 |

---

> ### Author Response · Authors · 2025-11-20
>
> >### **Response to Weakness 9**
>
> - **Inference throughput:**
>
> We present the inference overhead of WHT in the table below, which reports the throughput (tokens/sec) of the LLaMA-3.2-3B 4-bit model on a single A100 80GB GPU. The evaluation uses a prefill length of 2048 and a generation length of 64 at batch size 128. We compare the FP16 pre-trained model, a quantized model with LoRA (corresponding to CLoQ), a sparse adapter (SHiRA), and FT-based adapters including WHA (QWHA), DCA (LoCA), and DHA (SSH).
>
> As stated in SHiRA [1], sparse adapters provide a slight speedup over LoRA due to their simple scatter operations, compared with the low-rank matrix multiplications in LoRA. WHA introduces an additional WHT operation, but because it is implemented with a fast Hadamard kernel, the associated overhead remains small: only a 1.9 percent throughput reduction relative to LoRA. In contrast, conventional FT-based adapters such as DCA and DHA incur substantial overhead, exhibiting a 50.9 percent drop in throughput compared to LoRA.
>
> Therefore, although QWHA applies an inverse WHT during inference, its overhead remains marginal, whereas other FT-based adapters suffer from significantly lower efficiency. We agree that including inference-time cost results improves the clarity of our efficiency discussion, and we have added these measurements in Table 19 of Appendix D.6.
>
> *Caption: Inference throughput (tokens/sec) of pretrained and quantized models with various adapters.*
>
> | Methods | Pre-trained | LoRA | Sparse | WHA | DCA/DHA |
> |---------|-------------|-------|--------|------|----------|
> | Throughput (tokens/sec) | 66.7 | 188.1 | 191.9 | 184.6 | 92.4 |
>
> - **Inference memory usage:**
>
> We also measured inference-time memory usage under the same conditions as the throughput evaluation. The results are shown in the table below.
>
> QWHA reduces overall memory usage by 13.0 percent compared to CLoQ, while both methods use 3.04 GB for model weights. This improvement is attributed to the sparse adapter structure and its efficient scatter operations. Moreover, because QWHA’s inverse WHT is implemented without heavy matrix multiplications, it introduces no extra memory overhead and reduces the peak memory footprint. As a result, QWHA achieves more than a 10 percent reduction in total memory consumption. We have also added this result in Table 20 of Appendix D.6.
>
>
> *Caption: Peak memory usage (GB) of each method.*
>
> | Methods | QWHA | CLoQ |
> |---------|------|-------|
> | Memory Usage (GB) | 52.68 | 59.53 |
>
> [1] Sparse High Rank Adapters, https://arxiv.org/abs/2406.13175.

---

> ### Author Response · Authors · 2025-11-20
>
> >### **Response to Question 1**
>
> As discussed in detail in **Response to Weakness 1**, the seemingly large scaling factor of  $  \alpha = 4000  $  in QWHA is not the result of hyperparameter tuning, but rather a correction for an implementation-specific scaling issue in the inverse WHT path that restores the intended effective scaling to approximately 1.0. In the revised paper, we explicitly clarify the distinction between  $ \alpha_{explicit} $  and  $ \alpha_{effective} $  and plan to update the implementation so that the provided scaling factor is used directly, without unintended division by  $ d_{in} $ .
>
> To verify that this large numerical  $  \alpha  $  does not influence the training process in an unintended way, we compared training loss curves, gradient norms, and convergence speeds for both QWHA and CLoQ. As shown in Figure 10 of the revised paper, QWHA exhibits training dynamics that closely match those of CLoQ: loss trajectories are similar, gradient norms remain on a comparable scale, and convergence progresses at a similar rate. This is expected, since both methods ultimately operate under an effective scaling factor near 1.0. Therefore, QWHA’s performance improvements arise from its adapter design and quantization-aware initialization, rather than from a favorable choice of  $  \alpha  $ .

---

> ### Author Response · Authors · 2025-11-20
>
> >### **Response to Question 2**
>
> As discussed in detail in **Response to Weakness 4**, the initialization scheme provides a strong accuracy boost across all adapters, which naturally makes the performance gap between different transform kernels appear smaller when AdaAlloc is applied uniformly. We agree that, in terms of accuracy alone, the initialization scheme plays a major role.
>
> However, even under the same initialization, WHA consistently offers a stronger foundation for reconstructing quantization errors. In several settings, DCA + AdaAlloc and DHA + AdaAlloc still fall short of WHA, and in some cases even drop below the low-rank CLoQ baseline (for example, on 3-bit GSM8k with LLaMA-3.2-3B). This suggests that while AdaAlloc is effective, other Fourier transforms cannot fully leverage it because their coefficient distributions are less effective to approximate with the sparse adapter structure. In contrast, the energy-concentrative behavior of the WHT allows AdaAlloc to operate at its full potential.
>
> Moreover, the impact of the transform kernel extends beyond accuracy. WHA provides substantially better training efficiency due to its fast Hadamard kernel, which is not the case for DCT- or DHT-based adapters. As a result, WHA achieves both stronger accuracy and superior efficiency under the same initialization scheme.

---

> ### Author Response · Authors · 2025-11-20
>
> >### **Response to Question 3**
>
> As discussed in **Response to Weakness 6**, we conducted an additional ablation study to directly evaluate whether introducing a second transform offers any accuracy benefit. The results compare 1D WHA and 2D WHA across 4-bit, 3-bit, and 2-bit settings on both CSQA and GSM8k.
>
> We find that the accuracy difference between 1D and 2D WHA is consistently small across nearly all conditions, indicating that adding a second transform does not meaningfully improve representational effectiveness in the QA-PEFT setting. While 2D WHA can be viewed as a natural variant of QWHA and applies AdaAlloc in the same manner, its accuracy benefits remain marginal.
>
> As a result, our design choice for WHA dimensionality is 1D, which provides the desired balance of accuracy and efficiency.
>
> *Caption: Benchmark accuracy (%) according to the transform dimensionality in QWHA in LLaMA-3.2-3B.*
>
> | Method | QA init. | Coeff Selection | Refine. | Adapter | 4bit |  | 3bit |  | 2bit |  |
> |---|---|---|---|---|---|---|---|---|---|---|
> |     |     |     |     |     | CSQA | GSM8k | CSQA | GSM8k | CSQA | GSM8k |
> | QWHA | O | AdaAlloc | O | 1D WHA | 66.11 | 41.47 | 64.8 | 39.58 | 57.03 | 29.11 |
> | QWHA | O | AdaAlloc | O | 2D WHA | 65.92 | 41.43 | 64.56 | 38.66 | 57.21 | 30.85 |

---

> ### Author Response · Authors · 2025-11-21
>
> >### **Response to Question 4**
>
> As discussed in detail in **Response to Weakness 1**, our goal in selecting the temperature parameter   $   t   $   was to ensure that the adapter maintains full-rankness for effective fine-tuning while still placing sufficient emphasis on high-error channels to maximize the impact of quantization-aware initialization. To this end, we conducted a sensitivity analysis on   $   t   $  , evaluating both its influence on parameter allocation (represented via normalized rank that reflects parameter-deprived channels) and its effect on final fine-tuned accuracy, which are presented below, respectively.
>
> *Caption: Normalized rank (%) of linear projections initialized with different temperatures on the 4-bit quantized LLaMA-3.2-3B model with   $   P(r=64)   $  .*
>
>
> | Temperature (t) | 0.25 | 0.5 | 1 | 1.5 | 2 |
> |-----------------|-----|-------|------|------|-------|
> | Query           | 99.87 | 99.87 | 99.86 | 99.83 | 99.79 |
> | Key             | 100.00 | 100.00 | 100.00 | 100.00 | 100.00 |
> | Value           | 100.00 | 100.00 | 100.00 | 100.00 | 100.00 |
> | Out             | 99.72 | 99.71 | 99.66 | 99.60 | 99.57 |
> | Up              | 100.00 | 100.00 | 100.00 | 100.00 | 100.00 |
> | Gate            | 100.00 | 100.00 | 100.00 | 100.00 | 100.00 |
> | Down            | 100.00 | 100.00 | 100.00 | 100.00 | 100.00 |
>
>
> *Caption: GSM8k Accuracy (%) of QWHA models initialized with different temperature values on the 4-bit quantized LLaMA-3.2-3B model with   $   P(r=64)   $  .*
>
> | Temperature | 0.25 | 0.5 | 1 | 1.5 | 2 |
> |-------------|------|------|------|------|------|
> | GSM8k Acc   | 40.11 | 41.39 | 41.47 | 40.64 | 40.04 |
>
> Our results show that QWHA is highly robust for   $   t \in [0.5, 1.0]   $  , where both the adapter rank and GSM8k accuracy remain stable with only marginal variation (within 0.08%p in accuracy). This indicates that allocation sharpness and downstream performance are not particularly sensitive within this range. Based on these findings, we adopt   $   t = 1   $   as the default setting, as it naturally prioritizes outlier-affected, large-magnitude channels while maintaining stable empirical performance and keeping the method simple to use in practice.

---

> ### Author Response · Authors · 2025-11-21
>
> >### **Response to Question 5**
>
> As detailed in **Response to Weakness 3**, we conducted additional experiments to assess the stability of QWHA under different random seeds. Beyond the original seed (42) reported in the paper, we fine-tuned LLaMA-3.2-3B using multiple new seeds and measured the accuracy differences relative to the original run. Across all bit-widths, the observed fluctuations are small, and importantly, QWHA consistently remains above all baselines even under the largest variations. These results indicate that the improvements provided by QWHA are stable and robust across different random seeds.
>
> *Caption: Accuracy differences from the original run across different random seeds for QWHA fine-tuning on LLaMA-3.2-3B. Standard deviation is denoted as Std.*
> | Bit-width | 1111   | 2222   | 3333   | 4444   | 5555   | Std. |
> |-----|--------|--------|--------|--------|--------|-------------|
> | 2   | -0.130 | 0.046  | -0.040 | -0.084 | -0.050 | 0.065 |
> | 3   | 0.177  | -0.054 | 0.057  | 0.290  | 0.006  | 0.138 |
> | 4   | -0.081 | 0.017  | -0.050 | 0.077  | -0.011 | 0.061 |

---

> ### Comment · Reviewer_L6oQ · 2025-11-26
>
> Thank you for the author's detailed reply; I have raised my score to 6.

---

> ### Author Response · Authors · 2025-11-26
>
> Dear reviewer L6oQ.
>
> We sincerely appreciate your careful feedback on our work, as well as the positive assessment and score raise regarding our additional responses and the updated manuscript.

---

### Official Review · Reviewer_RPse · 2025-11-04

**Soundness:** 3
**Presentation:** 3
**Contribution:** 3
**Rating:** 6
**Confidence:** 3

**Summary:**

The paper presents QWHA (Quantization-Aware Walsh-Hadamard Adaptation), a novel approach for enhancing fine-tuning efficiency in LLMs under low-bit quantization settings. By integrating Walsh-Hadamard transforms within the quantization-aware training framework, the authors aim to mitigate quantization-related errors, thereby improving model performance without substantial computational overhead. The experiments conducted on several benchmarks substantiate the proposed method's effectiveness, indicating its potential to advance the state-of-the-art in efficient model adaptation.

**Strengths:**

1. The introduction of a Walsh-Hadamard Transform (WHT) presents a robust method for mitigating quantization error, enhancing the representational capacity of the model while maintaining computational efficiency.
2. The authors provide thorough experimental validation using different models and a range of quantization levels, supporting the claims of improved performance and efficiency.

**Weaknesses:**

1. The experimental evaluation relies heavily on the Alpaca dataset, primarily focusing on instruction-following tasks. This raises questions about the generalizability of the approach to other domains or more complex datasets, such as those involving natural language understanding or generation tasks.
2. The dependency on specific hyperparameter configurations (e.g., the temperature parameter in AdaAlloc) is only marginally discussed. There is a lack of sensitivity analyses that would indicate how variations in hyperparameters impact model performance and error rates.

**Questions:**

1. How does QWHA perform on diverse datasets that extend beyond instruction-following tasks? Can you provide insights into its performance on datasets related to summarization or question-answering?
2. Could the authors elaborate on the decision-making process for hyperparameter settings, particularly the temperature parameter in AdaAlloc? Have you explored the performance implications of varying these settings?

---

> ### Author Response · Authors · 2025-11-20
>
> Thank you for your constructive comments. We appreciate the insightful questions and comments you have provided.
>
> >### **Response to Weakness 1 and Question 1**
>
> Regarding the generalizability of QWHA in generative tasks, we note that, in addition to the instruction-following dataset (Alpaca52k), **QWHA was also fine-tuned on the mathematical reasoning dataset GSM8k and evaluated on a generative benchmark using a zero-shot CoT question-answering GSM8k task.**
>
> As described in Section 4 (line 372 of the original paper), GSM8k was mentioned as: “*For arithmetic reasoning, we adopt the GSM8k dataset and evaluate zero-shot accuracy following [1].*” However, it was not explicitly stated that GSM8k was also used for fine-tuning as well as evaluation, which may have led to the misunderstanding that QWHA was trained only on Alpaca52k. We have clarified this point in the updated paper.
>
> Our experiments therefore extend beyond instruction-following tasks and demonstrate QWHA’s effectiveness across different domains. Specifically: (a) fine-tuning on Alpaca52k dataset evaluates general question-answering ability (e.g., CSQA scores), and (b) Fine-tuning on GSM8k dataset evaluates domain-specific mathematical reasoning and numerical understanding through zero-shot CoT reasoning.
>
> The results on GSM8k confirm that QWHA maintains strong reasoning performance even in domains that require multi-step problem-solving and mathematical comprehension, demonstrating that QWHA generalizes effectively beyond instruction-following datasets.
>
> [1] Training Verifiers to Solve Math Word Problems, https://arxiv.org/abs/2110.14168.

---

> ### Author Response · Authors · 2025-11-20
>
> >### **Response to Weakness 2 and Question 2**
>
> Regarding the choice of the temperature parameter $ t $, we aimed to select a value that preserves the full-rankness of the adapter weights for effective fine-tuning, while still being high enough to emphasize high-error channels and maximize the effectiveness of quantization-aware initialization. We conducted additional experiments evaluating the sensitivity of QWHA to $ t $ in terms of both parameter allocation (full rankness) and fine-tuned accuracy, and found that $ t = 1 $ satisfies these criteria well.
>
> For parameter allocation, we measure the rank of the adapter after initialization, which reflects whether parameters are sufficiently distributed across channels without depriving any channel of the capacity needed for effective fine-tuning. As shown in the table below, the rank of the initialized QWHA weights remains consistently high across all projection types, with variations below 0.2%. This indicates that the resulting parameter allocation for effective fine-tuning remains robust as long as $ t $ is within a reasonable range.
>
> *Caption: Normalized rank (%) of linear projections initialized with different temperatures on the 4-bit quantized LLaMA-3.2-3B model with $ P(r=64) $.*
>
> | Temperature (t) | 0.25 | 0.5 | 1 | 1.5 | 2 |
> |-----------------|-------|------|------|-------|------|
> | Query           | 99.87 | 99.87 | 99.86 | 99.83 | 99.79 |
> | Key             | 100.00 | 100.00 | 100.00 | 100.00 | 100.00 |
> | Value           | 100.00 | 100.00 | 100.00 | 100.00 | 100.00 |
> | Out             | 99.72 | 99.71 | 99.66 | 99.60 | 99.57 |
> | Up              | 100.00 | 100.00 | 100.00 | 100.00 | 100.00 |
> | Gate            | 100.00 | 100.00 | 100.00 | 100.00 | 100.00 |
> | Down            | 100.00 | 100.00 | 100.00 | 100.00 | 100.00 |
>
> For fine-tuned model accuracy, the GSM8k results shown below (also included as Table 13 in Appendix D.3) exhibit very small variation: within 0.08%p for $ t \in [0.5, 1.0] $, and within 1.5%p across the entire tested range. We note that while low $ t $ spreads parameters too uniformly across channels, under-allocating to channels with large quantization error, high $ t $ over-concentrates parameters in high-error channels while neglecting important coefficients inside low-magnitude channels. As a result, both excessively high and excessively low temperatures lead to drops in accuracy.
>
> *Caption: GSM8k Accuracy (%) of QWHA models initialized with different temperature values on the 4-bit quantized LLaMA-3.2-3B model with $ P(r=64) $.*
>
> | Temperature | 0.25 | 0.5 | 1 | 1.5 | 2 |
> |-------------|------|------|------|------|------|
> | GSM8k Acc   | 40.11 | 41.39 | 41.47 | 40.64 | 40.04 |
>
> As a result, within the robust range of $ t \in [0.5, 1.0] $, we adopt $ t = 1 $ as the default setting, as it naturally prioritizes outlier-affected, high-magnitude channels while maintaining stable empirical performance.
>
> We acknowledge that the original submission omitted this ablation. We added the results and further explanations to Appendix D.3 of the revised paper for better clarity and visibility.

---

### Official Review · Reviewer_c9iU · 2025-11-08

**Soundness:** 3
**Presentation:** 3
**Contribution:** 3
**Rating:** 8
**Confidence:** 4

**Summary:**

This paper addresses the challenge of deploying Large Language Models (LLMs) efficiently by integrating parameter-efficient fine-tuning (PEFT) with low-bit quantization (QA-PEFT). The authors identify two key limitations in current methods: (1) the limited representational capacity of LoRA-based adapters and (2) the high computational cost and lack of quantization awareness in existing Fourier-related Transform (FT) based adapters. To overcome this, the paper proposes QWHA (Quantization-Aware Walsh-Hadamard Adaptation). This method introduces two main contributions: (1) WHA, a novel adapter that uses a single, computationally efficient Walsh-Hadamard Transform (WHT) kernel, which the authors show is full-rank and effective at capturing outlier-induced quantization errors. (2) A novel quantization-aware initialization scheme, featuring "AdaAlloc" for adaptive parameter selection and "Value Refinement" to optimize initial values, which is the first (to our knowledge) such scheme for FT-based adapters. Experiments show that QWHA outperforms existing QA-PEFT baselines, particularly in 2-bit and 3-bit settings, while achieving training speeds comparable to LoRA.

**Strengths:**

- Originality & Significance: The problem is well-motivated, sitting at the practical intersection of model compression and adaptation. The paper's primary originality lies in being the first to successfully design a quantization-aware initialization strategy specifically for an FT-based adapter. This bridges a critical gap, combining the superior representational power of FT-adapters with the error-mitigation necessary for low-bit quantization, which LoRA-based methods struggle with.

- Quality: The methodology is concrete and well-supported by a diverse set of analyses. The design choices for WHA (using WHT and a single transform) are justified through clear empirical evidence, including rank analysis, cumulative energy analysis, and outlier component coverage. The two-stage initialization algorithm (AdaAlloc and Refinement) is a non-trivial and well-reasoned approach to solving the complex sparse approximation problem.

- Clarity: The paper is clearly written, systematically identifying the limitations of prior work and presenting its solution. The figures and tables effectively support the authors' claims about why WHA is a suitable kernel and why the proposed initialization is effective.

**Weaknesses:**

- Unclear Ablation of Performance Gains: The main experimental comparison in Table 3 is potentially confounding. QWHA, which includes the proposed quantization-aware initialization (QA Init.), is compared against other FT-based adapters (LOCA, SSH) that do not have this initialization. It is unclear whether QWHA wins because: (A) the WHA kernel is intrinsically superior to DCA/DHA for this task, or (B) WHA is the only FT-based method benefiting from a powerful initialization. A fair comparison would require applying a comparable initialization to LOCA/SSH or ablating QWHA without its initialization.

- Missing Analysis of Initialization Cost: The paper highlights QWHA's efficiency by comparing its training time to CLoQ (Table 5), showing they are similar. However, this ignores the pre-computation cost of the QUANTIZATION-AWARE ADAPTER INITIALIZATION (Algorithm 1). This process involves gathering activations, SVD, and, most notably, solving a least-squares problem (with a matrix inverse) for every output channel of every adapted layer. This complex procedure is computationally non-trivial, whereas LoRA's initialization is virtually instantaneous. Without reporting this initialization time, the claim of "efficiency" is incomplete, as the total time-to-model (Initialization + Training) may be significantly higher for QWHA.

- Significant Performance Gap on Reasoning Tasks: While QWHA shows strong performance on CSQA, there is a very large performance degradation on the GSM8k benchmark (Table 3) compared to the 16-bit fine-tuned baseline. For instance, on LLaMA-3.1-8B, the 16-bit baseline achieves 59.74, while the 4-bit QWHA only reaches 56.10. This substantial drop is not fully addressed and may suggest that the proposed adaptation and error-compensation scheme struggles to preserve the complex reasoning capabilities of the model. Further exploration on other reasoning tasks (e.g., AIME, MATH) would be needed to verify if this is a fundamental limitation of the approach.

**Questions:**

- To fairly assess the novelty of the WHA kernel itself, could the authors provide an ablation study comparing WHA, DCA (LOCA), and DHA (SSH) under two conditions: (a) all using the same proposed QA initialization (AdaAlloc + Refinement) and (b) all using a simple baseline initialization (e.g., random selection, zero-init)? This would isolate the true benefit of the WHT kernel versus the initialization scheme.

- Could the authors please report the wall-clock time and memory overhead required for the QUANTIZATION-AWARE ADAPTER INITIALIZATION phase? How does the total deployment time (Initialization Time + Fine-Tuning Time) of QWHA compare to a method like CLoQ?

- Do the authors have an explanation for the significant performance gap between the 16-bit fine-tuned model and QWHA on the GSM8k reasoning task? Does this imply a potential limitation of the WHA adapter in capturing parameters crucial for mathematical reasoning, or is it a general challenge for low-bit QA-PEFT?

---

> ### Author Response · Authors · 2025-11-20
>
> Thank you for your constructive comments. We appreciate the insightful questions and feedback you have provided.
>
> >### **Response to Weakness 1 and Question 1**
>
> Regarding the attributions of performance gains, we would like to emphasize that Table 4 is designed to break down the contributions of (a) the adapter type and (b) the quantization-aware initialization or parameter-selection method, under controlled conditions where all other factors are kept identical.
>
> - **Effect of the adapter type:**
>
> When the quantization-aware initialization is fixed, the performance differences depending solely on the adapter type are shown in lines 442-444 of Table 4. All adapter types (e.g., DCA, DHA, Sparse, etc.) are evaluated using the same initialization strategy (AdaAlloc + Refinement), so the observed differences reflect the representational capability of each adapter. For example, under 2-bit quantization with GSM8k fine-tuning, we observe a 1.6-2.6% accuracy gain arising from the adapter type alone.
>
> - **Effect of the initialization method:**
>
>  Under the same WHA adapter setting, lines 438-442 of Table 4 illustrate performance differences resulting from alternative initialization strategies. Line 438 corresponds to the case without any initialization, while lines 439-441 report results using other parameter-selection approaches. These comparisons show the effectiveness of our quantization-aware initialization. At 2-bit quantization on GSM8k fine-tuning, our method achieves more than a 4.6% improvement over no initialization, and a 2.0-4.5% improvement over other parameter-selection strategies including those adopted in previous FT-based works (once our value assignment and refinement are applied equally across strategies).

---

> ### Author Response · Authors · 2025-11-20
>
> >### **Response to Weakness 2 and Question 2**
>
> - **Adapter initialization time**:
>
> Regarding the initialization time of the adapters, we present the comparison across CLoQ, SSH, and QWHA in the table below, where QWHA achieves initialization latency comparable to CLoQ. This table also appears in the updated paper Appendix D.6, Table 17 for a better clarity and visibility.
>
> *Caption: Initialization latency (hours) for each method and model under 4-bit quantization.*
>
> | Method | LLaMA-3.2-3B | LLaMA-3.1-8B | Mistral-7B-v0.3 |
> |---|---|---|---|
> | CLoQ | 0.58 | 1.14 | 1.26 |
> | SSH | 3.85 | 8.09 | 8.58 |
> | QWHA | 0.66 | 1.34 | 1.46 |
>
> As described in Section 3, the initialization process of QWHA consists of two steps: (a) gathering activations and quantization errors, and (b1) selecting parameters and refining their values by applying transforms and solving a channel-wise least-squares problem. Likewise, the initialization process of CLoQ also consists of two steps: (a) gathering activations and quantization errors, and (b2) applying SVD followed by solving the least-squares problem (as described on page 6, under Equation (5) of CLoQ [1]).
>
> Although FT-based adapters involve additional matrix projections, the fast WHT kernel used in step (b1) enables QWHA to maintain overall initialization time comparable to that of CLoQ. In contrast, other FT-based adapters incur higher initialization latency due to slower transform operations. Furthermore, the initialization time is significantly smaller than the total fine-tuning duration. Therefore, QWHA maintains comparable end-to-end latency to CLoQ.
>
> - **Memory usage during the initialization**:
>
> The memory usage of each adapter initialization process is shown in the table below, where QWHA exhibits slightly lower memory usage compared to CLoQ. This table also appears in the updated paper Appendix D.6, Table 18 for a better clarity and visibility.
>
> *Caption: Memory usage (GB) and its breakdown in the LLaMA-3.2-3B 4-bit quantized model.*
>
> | Method | Total | Model Weight | Cached Hessian |
> |---|---|---|---|
> | CLoQ | 12.77 | 3.04 | 7.10 |
> | SSH | 11.93 | 3.04 | 7.10 |
> | QWHA | 11.52 | 3.04 | 7.10 |
>
> QWHA slightly reduces memory consumption by 1.2 GB compared to CLoQ as it avoids the memory-intensive SVD operation and instead performs lightweight transforms using the WHT kernel, whose tiled transforms are more memory-efficient. As a result, QWHA incurs lower memory overhead during initialization.
>
> [1] *CLoQ: Enhancing Fine-Tuning of Quantized LLMs via Calibrated LoRA Initialization*, https://arxiv.org/pdf/2501.18475

---

> ### Author Response · Authors · 2025-11-20
>
> >### **Response to Weakness 3 and Question 3**
>
> Regarding the accuracy drop compared to the full fine-tuned models, particularly on the GSM8k dataset, we would like to highlight that this is a general challenge for low-bit QA-PEFT, and QWHA effectively mitigates this gap compared to other methods.
>
> As noted, the models in Table 3 labeled as fully fine-tuned correspond to (a) high-precision (FP16) models that are (b) fully fine-tuned across all parameters. In contrast, QWHA, as well as all other QA-PEFT methods, starts from (a’) a quantized model, which already exhibits lower performance than the original FP16 model, and (b’) performs parameter-efficient fine-tuning (PEFT), where only a small subset of parameters (e.g., in low-rank or sparse adapters) are updated. Therefore, the accuracy gap between the fully fine-tuned models and QWHA reflects a fundamental limitation shared by all low-bit QA-PEFT methods, rather than an issue unique to our approach. This gap tends to be more pronounced on challenging reasoning tasks such as GSM8k, which are particularly sensitive to quantization errors and limited parameter capacity.
>
> A similar trend is also observed in prior work, for instance, Table 2 of CLoQ [1] shows that, for both LLaMA-3 8B and Mistral-7B, the GSM8k accuracy of quantized INT2 models fine-tuned with CLoQ is 2-7% lower than that of FP16 + LoRA models. While QWHA may still exhibit a performance drop compared to full fine-tuning, our adapter design and quantization-aware initialization effectively mitigate this degradation, resulting in a smaller accuracy gap than other QA-PEFT methods under the same quantization setting.

---

### Author Response · Authors · 2025-11-20
**Global Response to the Reviewers**

We sincerely thank all reviewers for their thoughtful feedback and insightful questions. Several themes appeared consistently across the reviews, so we address two of the most frequently raised concerns here:   (a) the diversity of the fine-tuning datasets and evaluation benchmarks beyond instruction-following Alpaca52k dataset, and (b) the ablation on the temperature parameter $t$ used in AdaAlloc.

>### **Fine-tuning Datasets and Evaluation Benchmarks**

Across multiple reviews, there was concern that QWHA may have been evaluated primarily on instruction-following data (Alpaca52k), raising questions about its generalizability to other domains.

To clarify, **QWHA was fine-tuned not only on Alpaca52k but also on the GSM8k mathematical reasoning dataset, and evaluated with generative zero-shot CoT tasks on GSM8k.** We acknowledge that the original paper did not clearly state that GSM8k was also used for fine-tuning as well as evaluation. This has been corrected in the updated paper.

By fine-tuning and evaluating the model on two distinct datasets: Alpaca52k for general instruction following and common-sense question answering and GSM8k for mathematical reasoning and generative multi-step numerical problem solving, our experiments show that QWHA generalizes well across both **general knowledge question-answering** and **domain-specific reasoning** tasks. In particular, the strong GSM8k results confirm that QWHA maintains robust reasoning capability even under low-bit configurations.

>### **Ablation on the Temperature $t$ in AdaAlloc**

Several reviewers asked whether the temperature parameter $t$ affects adapter performance and why $t = 1$ is used in our main experiments.

We conducted an ablation study on the effect of $t$ on fine-tuned model accuracy, summarized in the table below (also included as Table 13 in Appendix D.3). The results show that accuracy remains stable for $ t \in [0.5, 1.0] $, with only mild degradation at both very low and very high temperatures. This behavior is expected: a low $t$ distributes parameters too uniformly across channels, under-allocating to channels with large quantization error, while a high $t$ over-concentrates parameters in a few high-error channels and neglects important coefficients in intermediate-magnitude channels.

*Caption: GSM8k Accuracy (%) of QWHA models initialized with different temperature values on the 4-bit quantized LLaMA-3.2-3B model with  P(r=64) .*

| Temperature | 0.25 | 0.5 | 1.0 | 1.5 | 2.0 |
|-------------|------|------|------|------|------|
| GSM8k Acc   | 40.11 | 41.39 | 41.47 | 40.64 | 40.04 |

Regarding the choice of the temperature parameter $ t $, we aimed to select a value that preserves the full-rankness of the adapter weights for effective fine-tuning, while still being high enough to emphasize high-error channels in order to maximize the effectiveness of quantization-aware initialization. Through experiments, we determined that the adapter rank is also highly robust within $ t \in [0.5, 1.0] $. Therefore, we adopt $ t = 1 $ as the default setting, as it naturally prioritizes outlier-affected, high-magnitude channels while maintaining stable empirical performance.

We acknowledge that the original submission did not include this ablation.  We added the results and further explanations to Appendix D.3 of the revised paper for better clarity and visibility.

---

We thank the reviewers again for their constructive feedback.

---

### Author Response · Authors · 2025-12-02
**Summary of Rebuttal**

Dear Area Chair,

We truly appreciate your time and attention in reviewing our paper and rebuttal materials. Below is a concise summary of our communication with the reviewers regarding the most frequently asked questions and impactful aspects of our work during the rebuttal period, as well as the score update.

> **Score Update**

Before November 27, 12:00 (UTC), one reviewer raised their score from **4 to 6**, changing the overall scores from **8–6–4–8** to **8–6–6–8**. This update followed our additional responses and the updated manuscript, which addressed their technical concerns and clarified key points of the work. We confirm that, to the best of our knowledge, no non-academic factors were involved.

> **Major Points of Rebuttal**

1. We clarified the misunderstanding regarding the datasets used. Some phrasing in our original submission led reviewers to believe we fine-tuned only on a single dataset and evaluated solely on general knowledge QA benchmarks. We clarified that we used multiple datasets for fine-tuning and evaluation. In particular, we also fine-tuned on GSM8k, which evaluates arithmetic reasoning and multi-step problem solving. This clarification has been incorporated in the updated manuscript.

2. We added an ablation study on the temperature hyperparameter $ t $, which is used during the parameter selection stage of our FT-based sparse adapter (WHA). Our experiments show that accuracy remains stable within the reasonable range $ [0.5, 1.0] $, and we selected $ t = 1 $ according to our strategy for reducing quantization error after quantization. This result has been added to the revised paper.

3. We provided additional explanations and measurements regarding memory usage and latency, covering both (a) adapter initialization and (b) model inference. We clarified that WHA, by design of its binary components, has resource overhead comparable to LoRA, unlike other FT-based adapters that incur significant overhead.

We remain grateful for the constructive feedback provided by all reviewers and for your oversight throughout this process.

---

### Meta-Review · Area_Chair_BgVT · 2026-01-07

**Summary:**

The paper introduces QWHA, a parameter-efficient fine-tuning (PEFT) method designed for low-bit quantized LLMs. It addresses limitations in existing methods by proposing a WHA Adapter, together with a novel adapter initialization scheme incorporating adaptive parameter selection and value refinement. Experiments show consistent superiority over baselines, particularly in challenging 2-bit and 3-bit settings.

**Reviewer Concerns:**

Reviewers generally view the paper positively, praising the novelty of the initialization scheme and the theoretical justification for WHT. However, there are concerns regarding the fairness of baseline comparisons (isolating the effect of the kernel vs. the initialization), the computational cost of the initialization phase, and the lack of statistical rigor (single runs).

**Reviewer Scores:**

After rebuttal discussion, all reviewers vote for acceptance, the authors should note the issues raised in the discussions, and update accordingly in the refined version

---

### Decision · Program_Chairs · 2026-01-26

Accept (Poster)